# Molecular asymmetry in the cephalochordate embryo revealed by single-blastomere transcriptome profiling

**Che-Yi Lin**[1], **Mei-Yeh Jade Lu**[2], **Jia-Xing Yue**[3], **Kun-Lung Li**[1], **Yann Le Pétillon**[1], **Luok Wen Yong**[1], **Yi-Hua Chen**[2], **Fu-Yu Tsai**[1,4], **Yu-Feng Lyu**[1], **Cheng-Yi Chen**[1], **Sheng-Ping L. Hwang**[1], **Yi-Hsien Su**[1]*, **Jr-Kai Yu**[1,5]*

**1** Institute of Cellular and Organismic Biology, Academia Sinica, Taipei, Taiwan, **2** Biodiversity Research Center, Academia Sinica, Taipei, Taiwan, **3** State Key Laboratory of Oncology in South China, Collaborative Innovation Center for Cancer Medicine, Sun Yat-sen University Cancer Center, Guangzhou, China, **4** Department of Life Science, National Taiwan University, Taipei, Taiwan, **5** Marine Research Station, Institute of Cellular and Organismic Biology, Academia Sinica, Yilan, Taiwan

* yhsu@gate.sinica.edu.tw (Y-HS); jkyu@gate.sinica.edu.tw (J-KY)

**Data Availability Statement:** The raw RNA-seq data for 24 amphioxus blastomere samples have been deposited in the NCBI database under

## Abstract

Studies in various animals have shown that asymmetrically localized maternal transcripts play important roles in axial patterning and cell fate specification in early embryos. However, comprehensive analyses of the maternal transcriptomes with spatial information are scarce and limited to a handful of model organisms. In cephalochordates (amphioxus), an early branching chordate group, maternal transcripts of germline determinants form a compact granule that is inherited by a single blastomere during cleavage stages. Further blastomere separation experiments suggest that other transcripts associated with the granule are likely responsible for organizing the posterior structure in amphioxus; however, the identities of these determinants remain unknown. In this study, we used high-throughput RNA sequencing of separated blastomeres to examine asymmetrically localized transcripts in two-cell and eight-cell stage embryos of the amphioxus *Branchiostoma floridae*. We identified 111 and 391 differentially enriched transcripts at the 2-cell stage and the 8-cell stage, respectively, and used *in situ* hybridization to validate the spatial distribution patterns for a subset of these transcripts. The identified transcripts could be categorized into two major groups: (1) vegetal tier/germ granule-enriched and (2) animal tier/anterior-enriched transcripts. Using zebrafish as a surrogate model system, we showed that overexpression of one animal tier/anterior-localized amphioxus transcript, *zfp665*, causes a dorsalization/anteriorization phenotype in zebrafish embryos by downregulating the expression of the ventral gene, *eve1*, suggesting a potential function of *zfp665* in early axial patterning. Our results provide a global transcriptomic blueprint for early-stage amphioxus embryos. This dataset represents a rich platform to guide future characterization of molecular players in early amphioxus development and to elucidate conservation and divergence of developmental programs during chordate evolution.

accession number PRJNA556757 (https://www.
ncbi.nlm.nih.gov/bioproject/PRJNA556757).

**Funding:** This work was supported by the
Academia Sinica (https://www.sinica.edu.tw/en)
intramural fund to JKY and YHS, and by the
Ministry of Science and Technology, Taiwan, under
the grants MOST-102-2311-B-001-011-MY3 and
MOST-105-2628-B-001-003-MY3 to JKY, and
MOST-107-2321-B-001-017 to YHS. The funders
had no role in study design, data collection and
analysis, decision to publish, or preparation of the
manuscript.

**Competing interests:** The authors have declared
that no competing interests exist.

## Author summary

Studies in model animals have shown that asymmetrically localized maternal transcripts
play important roles in axial patterning and cell fate specification in early embryos. How-
ever, comprehensive analyses of the maternal transcriptomes with spatial information are
limited to a handful of organisms. Here, we use a PCR-based single-cell RNA sequencing
approach on separated blastomeres from the cleavage stage embryos of amphioxus, an
early-branching chordate. We generate a spatial transcriptomic map and demonstrate the
mosaic property of the early amphioxus embryo, contrasting to the general notion that its
embryogenesis is highly regulative. Our cross-species experiments further provide evi-
dence to support that some of the identified factors play functions in early axial pattern-
ing. Our results provide a global transcriptomic blueprint that can serve as a rich platform
for guiding future characterization of molecular players in early amphioxus development
and for comparative studies on testing conservation and divergence of developmental
programs during chordate evolution.

## Introduction

A key question in developmental biology is how different cell fates and body axes are specified
in early embryos. Based on experimental embryology studies, specification in early embryo-
genesis has traditionally been ascribed to one of two mechanisms, either 'mosaic' or 'regulative'
development [1]. Snails are considered to undergo mosaic development because loss of early
blastomeres causes the loss of later developing structures. On the other hand, sea urchin
embryos are thought to develop in a regulative manner because blastomeres separated at the
4-cell stage will develop into four small, but otherwise normal, pluteus larvae. Modern molecu-
lar analyses have demonstrated that maternally localized cytoplasmic determinants confer
"mosaic" properties to the blastomeres that inherit them, while cell-cell interactions via extra-
cellular signals result in "regulative" development [2]. These cytoplasmic determinants and sig-
naling pathways then activate downstream genes to specify different cell fates and embryonic
polarities.

More recent studies on a variety of animals have revealed that mosaic and regulative devel-
opmental models simply represent two extremes of a scale, and embryos can simultaneously
exhibit both attributes in interlocked components of complex gene regulatory networks. For
example, in the *Xenopus* embryo, the dorsal determinant *wnt11*, which encodes a paracrine
factor, is translocated from the vegetal to the dorsal side, where it stabilizes β-catenin transcrip-
tion factor [3]. These factors provide the initial driving force behind the formation of the
embryonic polarity, which later induces the Spemann-Mangold organizer that patterns the
whole embryo. Similarly, in zebrafish embryos, the maternal factors that control dorsal-ventral
patterning are asymmetrically localized along the animal-vegetal axis after fertilization. Vegetal
pole-enriched transcripts, such as *wnt8a*, are transported by kinesin motor proteins along
microtubules to the dorsal side [4,5]. Subsequently, maternal β-catenin is stabilized and turns
on nodal signaling to promote dorsal cell fate specification. In addition to wnt ligands, Hwa, a
maternal protein localized on the plasma membrane of dorsal blastomeres, protects β-catenin
from degradation and is an essential determinant for the formation of the dorsal organizer in
zebrafish and *Xenopus* embryos [6]. Maternal transcripts enriched in the animal hemisphere
are known to include *pou2*, *radar* and *runx2bt2*, which function upstream of *bmp* and the *vox/
vent* gene family to promote the ventral cell fate specification [4]. Although many maternal
determinants have been identified in vertebrate model systems, the extent to which a particular

species relies on mosaic and regulative mechanisms for its embryonic cell fate specification can only be delineated by a systematic analysis of developmental programs.

Outside the vertebrate lineage, the invertebrate chordate tunicate embryos are considered to be highly mosaic, although formation of several cell types requires inductive signals as well. Microarray and whole-mount *in situ* hybridization (WMISH) analyses in the tunicate *Ciona intestinalis* have identified a group of asymmetrically localized maternal transcripts during early cleavage stages [7–9]. Among these transcripts, about 50 are localized to the posterior-vegetal cytoplasm/cortex and play important roles in several developmental processes, including determination of the anterior-posterior axis, muscle cell fate, and germ cells. The microarray data also identified 25 gene products that are predominant in the animal pole, although the localization of these transcripts was not confirmed by WMISH. A lack of strong anterior determinants would be consistent with the observation that removal of the anterior cytoplasm has no effect on tunicate embryogenesis [10]. Thus, these systematic studies suggest that transcriptional determinants of the early body plan in tunicate embryos are at least mostly localized to the posterior vegetal region to promote the posterior fate, while the lack of posterior-vegetal determinants in the anterior blastomeres results in default anterior cell fate specification [11].

Unlike tunicates, amphioxus, another invertebrate chordate, has long been thought to employ a mostly regulative mechanism for its early development [12,13]. Nevertheless, classical vital dye-staining and blastomere separation experiments have revealed that amphioxus embryos exhibit some level of mosaic development during early cleavages [14,15]. For example, the anterior and posterior halves, separated along the second cleavage plane, of the 4-cell stage amphioxus embryo appear to have different developmental capacities. In most cases, one half will grow into a slightly abnormal larva containing complete embryonic structures, while the other will become a malformed embryo without a notochord, neural tube and somites [14]. Furthermore, when an embryo is separated along the third cleavage plane at the 8- or 16-cell stage, the animal half will initially form a ciliated ball and later become a flattened double-layered disk, sometimes with small amounts of mesodermal or endodermal tissues on the inside; meanwhile, the vegetal half will develop into a cell mass possessing certain features of the digestive tract and mesodermal tissues, but mostly lacking ectodermal tissues [14]. This mosaic aspect of development may be traced back to the fertilized egg [16], and is supported by the asymmetric distribution of *nodal* transcripts toward the animal hemisphere [17,18]. In addition, our previous studies showed that soon after fertilization, maternal transcripts of germline markers *vasa*, *nanos* and *piwil1* aggregate into a compact germ granule that is inherited by only one blastomere after the first and several subsequent cleavages [19,20]. As such, amphioxus blastomeres separated at the 2-cell stage have different developmental capacities, with one blastomere developing into a typical larva and the other one becoming a larva with a curly-tail phenotype with disorganized posterior structures; thus, the first two blastomeres are not identical, and certain asymmetrically localized maternal transcripts likely participate in specifying cell fates and body axes in early amphioxus embryos [19]. However, the identities of those maternal determinants remain mostly unknown, and it is unclear whether amphioxus utilize vertebrate-like determinants to trigger appropriate axial development [21,22].

In this study, we used a PCR-based single-cell RNA sequencing (scRNA-Seq) approach to systematically survey the distribution of transcripts at the 2-cell and 8-cell stage embryos of the amphioxus *Branchiostoma floridae*. Based on the scRNA-Seq data and further validations by WMISH, we identified two groups of differentially enriched transcripts: (1) vegetal tier/germ granule-enriched and (2) animal tier/anterior-enriched transcripts. Functional experiments using zebrafish as a surrogate model system suggested that the animal tier/anterior-enriched zinc finger protein, zfp665, may function in early dorsal-ventral patterning. Overall, our study

provides a global transcriptomic blueprint for guiding future functional studies in amphioxus and comparative studies on the evolution of developmental mechanisms in chordates.

## Results

### Sample preparation and quality control for RNA-seq

To systematically identify differentially enriched transcripts in amphioxus, we generated two sets of samples at two different embryonic stages (Fig 1A). The first set of samples was comprised of paired individual blastomeres that were separated from 2-cell stage embryos. These early blastomeres are morphologically indistinguishable. Our second set of samples included the animal and vegetal tiers separated at the 8-cell stage. At this stage, the blastomeres of the animal tier are smaller than those of the vegetal tier. Multiple pairs of samples derived from individual embryos were obtained for each stage and then subjected to reverse transcription and PCR using the Smart-seq2 method (Fig 1A) [23,24]. Our previous studies showed that the germ granule and its associated germline marker-encoding transcripts are segregated into one of the two blastomeres of the 2-cell stage embryo; the granule is later inherited by one of the vegetal blastomeres in the 8-cell stage embryo [19,20]. To identify presumptive germ granule-positive and -negative blastomeres, we performed qPCR to quantify transcript levels of four germline markers (*vasa*, *nanos*, *piwil1*, and *tdrd*/ID118019) [25]. Four ubiquitously expressed genes and two housekeeping genes were used as internal controls (S1 Fig). We first excluded unbalanced samples (6 out of 28 sample pairs) that exhibited a ΔCt, difference of cycle threshold (Ct), for β-actin over 2 between the two blastomeres, possibly due to RNA degradation or incomplete reverse transcription in one of the paired samples (S1 Fig). Secondly, we removed another 6 embryos that showed inconsistent trends among the four germline markers. In the remaining 16 embryos, 12 showed expected asymmetric enrichment of germline markers between the two blastomeres, and 6 of these embryo sets (12 samples) were selected for sequencing (Fig 1B) based on cDNA quality assessed by a Bioanalyzer. For the 8-cell stage samples, 7 out of the 9 sets with ΔCt of β-actin less than 2 showed the expected vegetal enrichment pattern of germline markers, and 6 of these (12 samples) were subjected to RNA sequencing (Fig 1F). Approximately 25.5 million reads (2x181 bp paired-end) were obtained for each sample.

The reads were mapped to the amphioxus *Branchiostoma floridae* reference genome [26]. For individual samples, the proportion of reads mapped to the genome ranged from 44% to 77%, and reads mapped to mitochondrial DNA ranged from 18% to 53% (S2A Fig). On average, 63% and 32% of the reads were mapped to the *B. floridae* genome and its mitochondrial DNA, respectively (S2B Fig). The proportion of reads mapped to mitochondrial DNA varied among samples, but this variation could not be attributed to their batch origins. Interestingly, we noticed that the proportion of reads mapped to mitochondrial DNA were slightly but consistently higher in germ granule-positive samples from both 2- and 8-cell stage embryo sets (paired *t*-test, *p* = 0.0064), indicating that more mitochondrial transcripts were produced in the germ granule-positive blastomeres. A high proportion of mitochondrial DNA-derived transcripts was also observed in the ESTs of an amphioxus cDNA database at the unfertilized egg stage (44.2%), in comparison to the other stages (from 9.8% to 26.7%) [27]. This phenomenon may be related to the mitochondrial aggregate that is associated with the germ plasm near the vegetal pole of amphioxus zygotes [19,28].

We then estimated the expression levels of each annotated transcript. Based on the averaged FPKM (fragments per kilobase per million) values, a total of 23,517 genes had detectable transcript levels, with 12,445 genes showing FPKM at least of 1 on average (S2C Fig). According to sequencing depth analysis, over 90% of the moderately expressed transcripts (FPKM > 1)

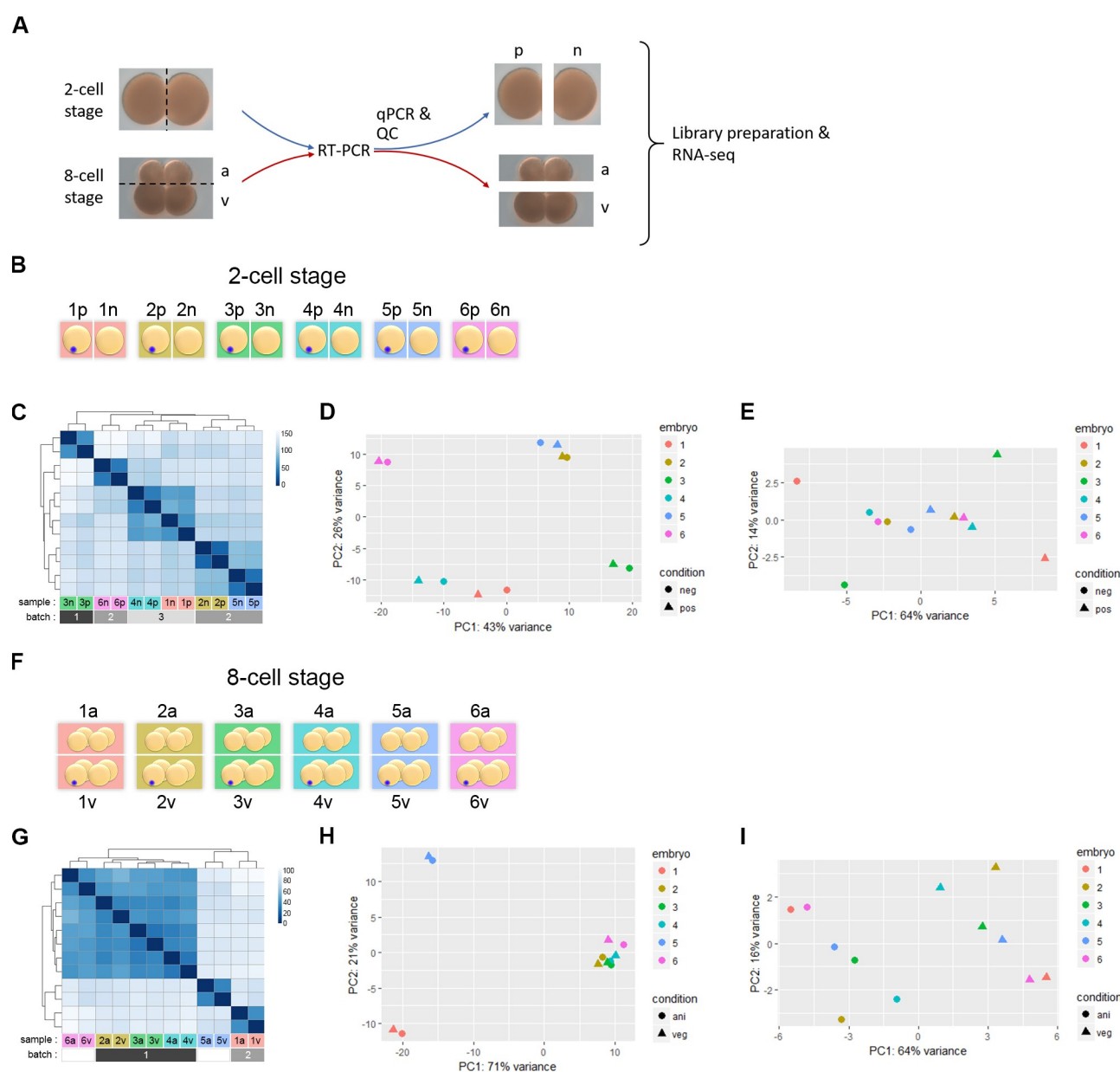

**Fig 1. RNA-seq analyses of blastomeres from the 2-cell stage and animal/vegetal tiers from the 8-cell stage.** (A) Images of 2-cell and 8-cell amphioxus embryos, with an illustration of the strategy for RNA-seq. Dashed lines indicate the planes separating blastomeres of each paired sample. The presumptive germ granule-positive (p) and germ granule-negative (n), and animal-tier (a) vs. vegetal-tier (v) blastomeres are indicated. (B) A schematic representation of the experiment using single blastomeres from 2-cell stage embryos (six biological replicates). Blue dot denotes the germ granule in the presumptive germ granule-positive blastomere in each embryo (from 1p to 6p) based on qPCR analysis (S1 Fig); the other cell from the same embryo is designated as the germ granule-negative cell (from 1n to 6n). (C) Hierarchically clustered heatmap based on the Euclidean distances between 2-cell stage samples. Identification of the samples and their batch of origin are shown at the bottom of the heatmap. (D) PCA of the twelve 2-cell stage samples shows strong embryo and batch effects. (E) PCA of the twelve 2-cell stage samples after data normalization to account for individual embryo differences. (F) A schematic representation of the experiment on animal (1a-6a) and vegetal tiers (1v-6v) from 8-cell stage embryos (six biological replicates). Blue dots denote the germ granule that is present in one of the blastomeres of the vegetal tier. (G) Hierarchically clustered heatmap based on the Euclidean distances between 8-cell stage samples. Identification of the samples and batch of origin are shown at the bottom of the heatmap. Batch identities of 5a/v and 6a/v were not recorded. (H) PCA of the twelve 8-cell stage samples shows strong embryo and batch effects. (I) PCA of the twelve 8-cell stage samples after data normalization to account for individual embryo differences.

could be detected when the counts reach 2 million (S2D and S2E Fig). Furthermore, outlier detection revealed comparable ranges of Cook's distances and no outliers across both data sets (S3A and S3B Fig). Estimation of dispersion values for expressed genes among biological replicates revealed that transcripts expressed at higher levels had smaller dispersion values than those with lower expression levels (S3C and S3D Fig), consistent with typical RNA-seq datasets [29]. In sum, these results showed that the sequencing depth and data quality for the 24 sequenced samples were appropriate for clustering analyses.

Both hierarchical clustering (Fig 1C and 1G) and principal component analysis (PCA, Fig 1D and 1H) consistently grouped samples derived from the same embryo together. Moreover, most embryo sets derived from the same batch were grouped as well. Therefore, our RNA-seq data showed strong individual and batch effects. To minimize individual and batch effects, we used the limma script (R package) and found that after normalization, the PC1 could explain the majority of variance and successfully segregate the 2-cell or 8-cell stage embryo samples according to their blastomere identities (Fig 1E and 1I). This result confirmed our qPCR-based assignment of the germ granule-positive and -negative samples and allowed us to further identify differentially enriched transcripts (DETs) between the two samples of each set.

## Identification and visualization of DETs

We employed two commonly used software packages (DESeq2 and edgeR) to identify DETs from the RNA-seq data. Analyses with an FDR cutoff of 0.1 revealed 121 DETs based on DESeq2 and 164 DETs by edgeR for the 2-cell stage embryos (Fig 2A and 2B; S1 Table). Comparing the DESeq2 and edgeR data, we found 106 DETs (69.3%) that were identified by both methods in the germ granule-positive cell (Fig 2G), while only five DETs (23.8%) were commonly identified in the germ granule-negative cell (Fig 2H). Between the 8-cell stage animal and vegetal tiers, we identified 501 and 571 DETs based on DESeq2 (Fig 2D) and edgeR (Fig 2E), respectively. Among them, 222 (60.5%) were commonly identified in the animal tier, and 169 transcripts (53.8%) were commonly found in the vegetal tier (Fig 2I and 2J). Overall, we identified 111 (106+5) transcripts that were differentially enriched in the germ granule-positive or -negative blastomere at the 2-cell stage, and 391 (222+169) transcripts that were differentially enriched in the animal or vegetal tier at the 8-cell stage. Notably, fold changes of low-level transcripts were smaller in the DESeq2 analyses (Fig 2A and 2D). Judging from FPKM distributions (Fig 2C and 2F), we found that edgeR identified more DETs with low-level expression than DESeq2. Therefore, DESeq2 appears to have higher stringency, while edgeR is more sensitive in detecting transcripts with low levels. Thus, in this manuscript, we chose to present the results from DESeq2 in the main figures and those from edgeR in supplementary figures.

To visualize the data, we generated four-parameter 2D scatter plots based on the analyses using DESeq2 (Fig 3A; S2 Data) and edgeR (S4 Fig; S3 Data). Overall, we found that most of the DETs fell into two groups. One group represents transcripts that are enriched in the animal tier (blue circles in Fig 3A). The other major group of DETs includes those in the presumptive germ granule-positive blastomere and/or the vegetal tier (yellow, red and orange circles in Fig 3A). Notably, only a few DETs (4 from DESeq2 and 15 from edgeR, green circles in Fig 3A, S4 Fig, respectively) were found in the presumptive germ granule-negative blastomere, and most of these had low FPKM values, suggesting that the presence of the germ granule is the predominant factor contributing to the molecular asymmetry at the two cell stage.

In order to validate our transcriptome analyses, we examined the 2D scatter plot for the positions of maternal transcripts that have previously reported distribution patterns in early

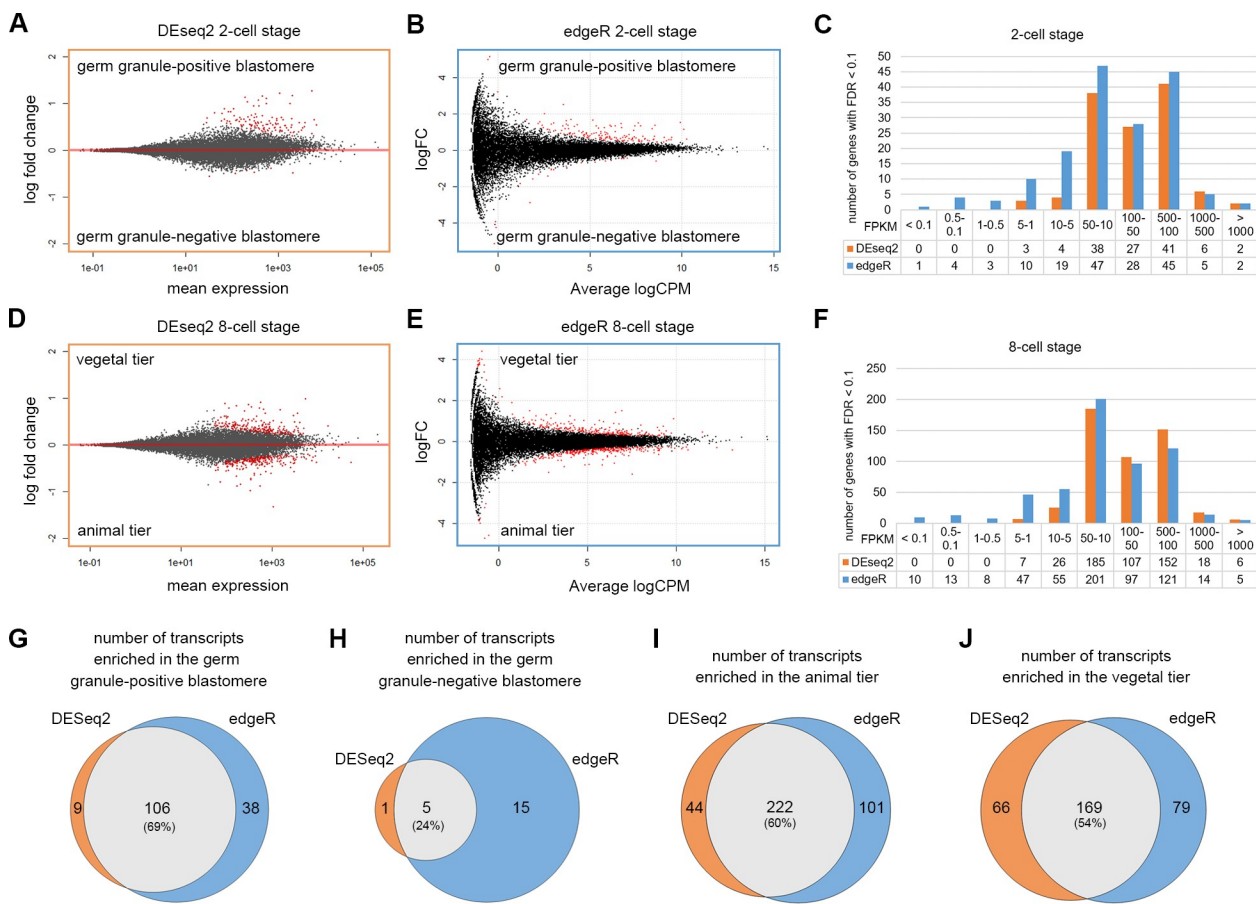

**Fig 2. Identification of differentially enriched transcripts (DETs).** (A-B) MA-plots showing the log2 fold changes between the germ granule-positive and negative cells over the average transcript level of each gene among the 2-cell stage samples analyzed by using DESeq2 (A) or edgeR (B). The DETs (FDR < 0.1) are shown in red. (C) Distribution of FPKM values of DETs based on DESeq2 (orange) and edgeR (blue) analyses at the 2-cell stage. (D-E) MA-plots showing the log2 fold changes between the animal and vegetal tiers over the average transcript level of each gene among the 8-cell stage samples analyzed by using DESeq2 (D) or edgeR (E). (F) Distribution of FPKM values of DETs based on DESeq2 (orange) and edgeR (blue) analyses at the 8-cell stage. (G-J) Venn diagrams show the number of DETs in the germ granule-positive cell (G), germ granule-negative cell (H), animal tier (I) and vegetal tier (J), comparing results between DESeq2 and edgeR analyses. Underlying data are available in S1 Data.

amphioxus embryos. *Tcf* and *nodal* transcripts are known to be enriched in the animal tier [17,18,30]; indeed, we found that both *Tcf* and *nodal* were located in the first quadrant, which corresponds to the germ granule-negative blastomere and the animal tier (blue in Fig 3B and 3C). Moreover, our RNA-seq data show that the fold change of *nodal* between animal and vegetal tier is smaller than that of *Tcf*, recapitulating previously reported *in situ* hybridization data showing that while *nodal* transcripts are enriched in the animal half, its transcripts are also present in parts of the vegetal tier [18]. Similarly, several previously characterized germ granule-associated transcripts in various amphioxus species, including *nanos*, *tdrd*, *SoxB1b*, *pl10* and *bruno2*, were found in the third quadrant, which represents the germ granule-positive blastomere and the vegetal tier (Fig 3B and 3C) [19,20,25,31,32]. Notably, two other germline markers, *vasa* and *piwil1*, were also found in the third quadrant, but the transcript levels were not significantly different between the paired samples. This may be due to the fact that diffuse signals for *vasa* and *piwil1* transcripts are also present throughout the entire embryo [19,20]. Overall, these results confirm that our RNA-seq analyses are valid for identifying DETs, and the results are consistent with previous knowledge.

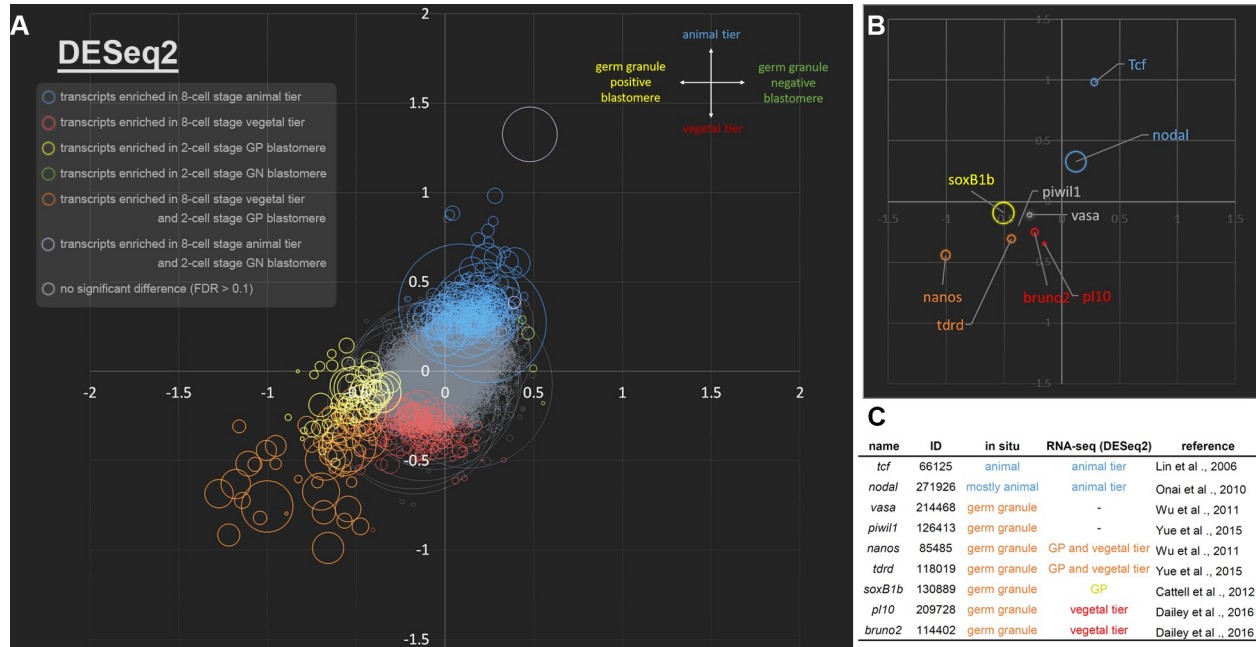

**Fig 3. Visualization of the transcriptome data showing asymmetric distributions of transcripts in the early amphioxus embryo.** (A) A scatter plot showing the log2 fold change of each gene between animal (positive y-axis) and vegetal tiers (negative y-axis) at the 8-cell stage, and its corresponding log2 fold change between germ granule-negative (positive x-axis) and positive blastomeres (negative x-axis) at the 2-cell stage (based on DESeq2). Size of the circle depicts FPKM values of the corresponding DET. Color of the circle indicates that the transcript is differentially enriched in a specific embryonic domain. Blue circles (total number = 264) are DETs in the animal tier; red circles (178) are DETs in the vegetal tier; yellow circles (58) are DETs in germ granule-positive blastomere; green circles (4) are DETs in germ granule-negative blastomere; orange circles (57) are DETs in both the germ granule-positive blastomere and the vegetal tier; purple circles (2) are DETs in both the germ granule-negative blastomere and the animal tier. Transcripts that are not enriched are in gray. (B) A scatter plot showing the spatial distributions of nine previously characterized transcripts. (C) Summary information for the nine previously characterized genes. GP, germ granule-positive blastomere. ID, gene model ID from genome assembly v2.0.

## Verification of newly identified enriched transcripts by whole-mount *in situ* hybridization

To validate previously unknown DETs identified in our RNA-seq data, we selected 23 transcripts that were enriched in both the germ granule-positive blastomere and the vegetal tier, and 10 transcripts enriched in either the germ granule-positive blastomere or the vegetal tier (S2 Table) for WMISH; the cDNA clones of these transcripts could be readily obtained from our existing EST library [27]. We successfully detected signals for 13 of the transcripts (Fig 4A and 4B), and showed that each was aggregated into a compact granule at the 2- and/or 8-cell stage (Fig 4C and 4D). In five cases (Fig 4Ca-c and 4Dj-k), the compact signal could not be detected until the 8-cell stage, possibly due to a limitation of the chromogenic detection method; the aggregated signal could be observed for two of the five transcripts at the 2-cell stage when fluorescence *in situ* hybridization was used (Fig 4Dj-k). It is also possible that some maternal transcripts may gradually become concentrated in the germ granule during the period from the 2- to 8-cell stage, or alternatively, zygotic gene activation between the 2- to 8-cell stage may generate transcripts that become associated with the granule (see further analysis below). To confirm whether this compact granule was the germ granule *per se*, we chose four transcripts to perform double fluorescence *in situ* hybridization and confirmed that they were co-localized with *nanos*, a known constituent of the germ granule (Fig 4D). In addition to the signals in the germ granule, transcripts of several genes that we examined displayed visible cytoplasmic distributions in diffused patterns (Fig 4Cc, 4Cf-i and 4Dm), similar to our

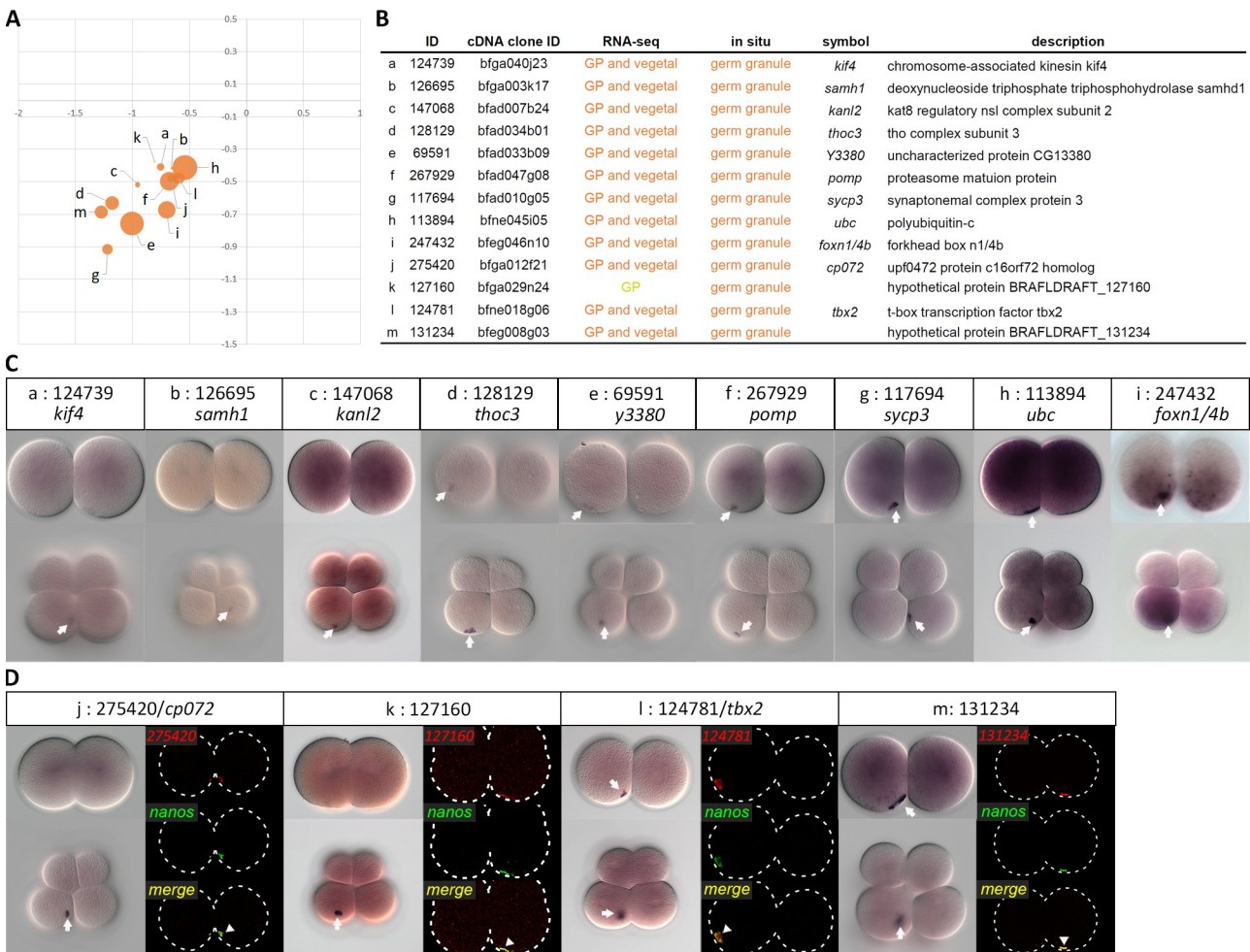

**Fig 4. Validation of newly identified germ granule-enriched transcripts by WMISH.** (A) Visualization map of 13 germ granule-enriched transcripts based on DESeq2. (B) Annotations of the validated transcripts. (C) WMISH of the corresponding transcripts at 2- and 8-cell stages. The white arrows indicate aggregated signals on the vegetal side. Animal pole is up and vegetal pole is down. (D) Double fluorescence *in situ* hybridization showed that four newly identified transcripts were co-localized with *nanos* transcripts in the germ granule. The white arrowheads indicate co-localized signals. The white arrows indicate the compact germ granule signals. Embryo outlines are demarcated by dashed lines. The WMISH images show representative expression patterns (> 80%, n ≥ 5).

previous observation for amphioxus *vasa* transcripts [19]. Furthermore, we observed *foxn1/4b* transcripts were asymmetrically localized to the vegetal hemisphere at the 2-cell stage and were more abundant in the germ granule-bearing blastomere at the 8-cell stage (Fig 4Ci).

Similar to amphioxus, several zebrafish germline markers are known to be aggregated into a compact granule at the early cleavage stage [33,34]. Thus, we proceeded to investigate whether orthologs of our newly discovered amphioxus germ granule-enriched transcripts also comprised aggregates in zebrafish embryos. Intriguingly, except *vasa*, which served as a positive control, six orthologs of the amphioxus germ granule-enriched transcripts that we tested were not aggregated into compact granules in zebrafish embryos at early cleavage stages (S5A Fig). To examine whether these transcripts are later enriched in zebrafish primordial germ cells after their migration to the developing gonad, we performed WMISH at 30 hpf, when *vasa*-positive cells were observed in the anterior end of the yolk extension (black arrow in S5B Fig). Much like the observations made at cleavage stages, orthologs of the six amphioxus germ

granule-enriched transcripts were not detected in the zebrafish developing gonads (S5B Fig). Therefore, except for a conserved core of germline determinants (such as *vasa*, *nanos*, and *piwi*), the sets of germ granule-enriched transcripts are probably variable among different animals.

From our EST library, we also obtained 22 cDNA clones of the animal tier-enriched transcripts for WMISH (S2 Table). We successfully detected signals for 10 of these transcripts (Fig 5A and 5B), and showed that nine (except naa38/279155) were clearly more concentrated

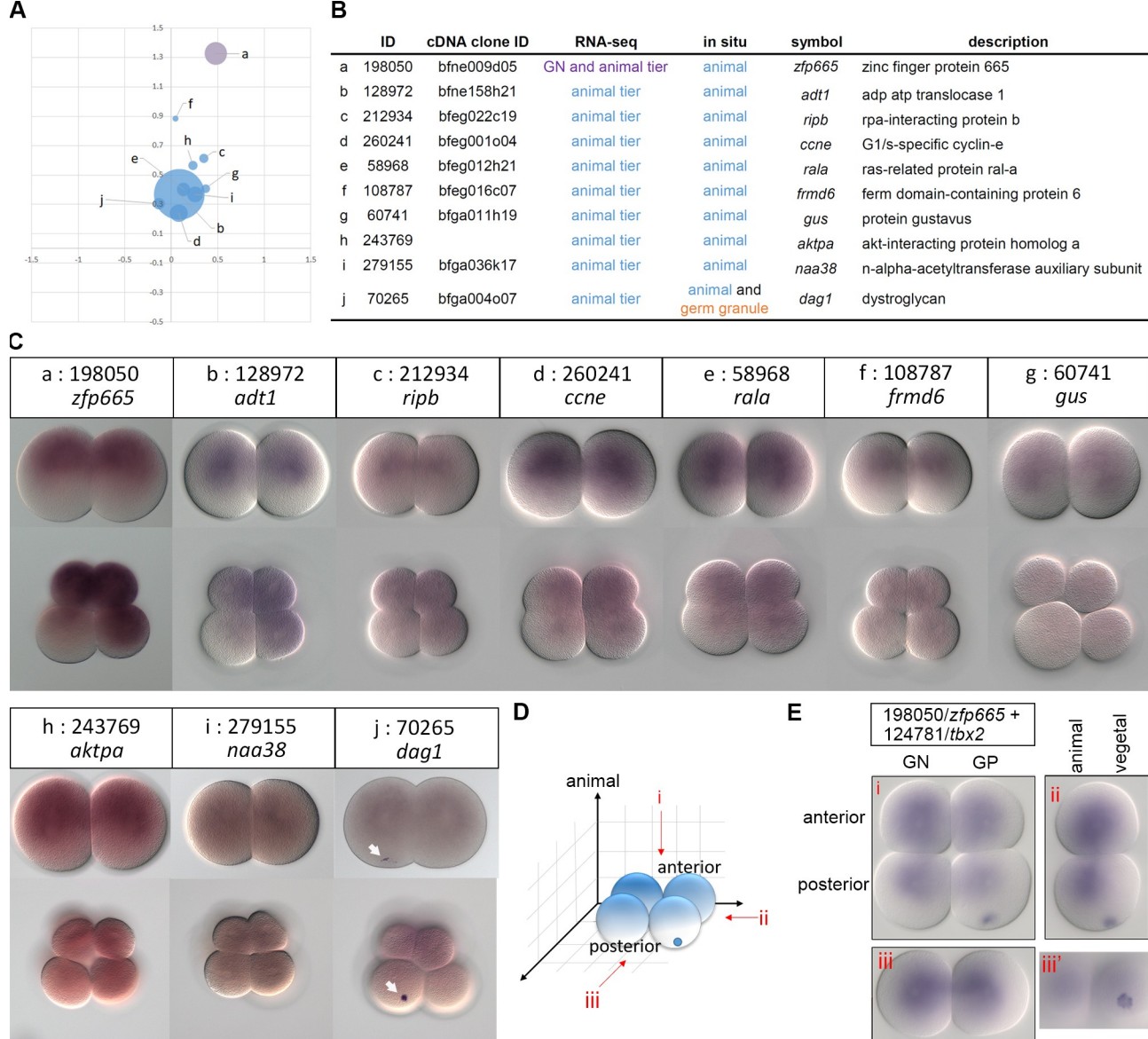

**Fig 5. Validation of newly identified animal tier-enriched transcripts by WMISH.** (A) The scatter plot shows spatial distributions of ten animal tier-enriched transcripts based on DESeq2. (B) Annotations of the validated transcripts. (C) WMISH of the corresponding transcripts at the 2- and 8-cell stages. The arrows indicate the aggregated signals on the vegetal side. Animal is up and vegetal is down. (D) A schematic representation showing different views of a 4-cell stage amphioxus embryo, (i) animal pole view, (ii) lateral view, and (iii) posterior view. Transcripts enriched in the germ granule and toward the animal pole are illustrated in blue. (E) Double WMISH of *zfp665* (198050) and *tbx2* (124781) at the 4-cell stage. Embryos are observed from different views as indicated in (D). A surface focal plane of the posterior view (iii') showing the germ granule-enriched transcripts. The WMISH images show representative expression patterns (> 80%, n ≥ 5).

toward the animal pole at the 2-cell stage (Fig 5C); at this stage, the animal pole could be recognized by the position of the polar body (S6 Fig). At the 8-cell stage, these transcripts were detected in all four animal blastomeres, but their distribution patterns in the vegetal blastomeres varied (Fig 5C). Besides being in the animal blastomeres, transcripts of *zfp665*/198050, *adt1*/128972 and *ripb*/212934 showed unequal distribution among the vegetal blastomeres (Fig 5Ca-c); *ccne*/260241 and *rala*/58968 transcripts were weakly detected in all vegetal blastomeres and were devoid near the vegetal pole (Fig 5Cd-e). For transcripts *frmd6*/108787, *gus*/60741, *aktpa*/243769 and naa38/279155, no clear patterns in the vegetal blastomeres can be observed at the 8-cell stage (Fig 5Cf-i). Interestingly, transcripts of *dag1*/70265 showed animal enrichment but were also aggregated into a compact granule in one vegetal blastomere (arrows, Fig 5Cj). Among the 10 selected transcripts, *zfp665*/198050 is the only one that showed enrichment in both the germ granule-negative blastomere and the animal tier, based on the RNA-seq data (Fig 5A and 5B). At the 8-cell stage, we observed that *zfp665*/198050 transcript signals were stronger in six of the eight cells and weaker in two of the vegetal cells (Fig 5Ca). The first and second cleavages divide the amphioxus embryo into left/right and anterior/posterior blastomeres, respectively, and the germ granule is always inherited by one of the two posterior blastomeres at the 4-cell stage [19]. Using *tbx2*/124781 transcripts as a landmark (known to aggregate in the germ granule, Fig 4Dm), we confirmed that *zfp665*/198050 transcripts were distributed toward the animal pole, more in the two anterior and the posterior germ granule-negative blastomeres, and less abundant in the germ granule-bearing blastomere (Fig 5D and 5E).

In summary, among the genes we selected for validation and successfully obtained WMISH signals, most of them displayed expected patterns, providing the first glimpse of molecular asymmetry in the early amphioxus embryo. Therefore, our genome-wide transcriptomic blueprint with spatially-resolved information may serve as a starting point for identifying previously unknown maternal transcripts that are functionally important for early embryogenesis.

## Gene ontology (GO) enrichment analysis suggests possible functions of DETs

To unbiasedly analyze the potential functions of asymmetrically distributed transcripts identified from our RNA-seq data, we performed separate GO enrichment analyses on the animal tier- and germ granule positive/vegetal tier-enriched transcripts. We found that most of the animal tier-enriched transcripts are annotated with the following GO terms: (1) negative regulation of mRNA metabolism, (2) cellular protein localization and (3) SUMO transferase activity (Fig 6A and 6B). For the germ granule-positive/vegetal tier-enriched transcripts, more biological processes and molecular functions were identified (Fig 6C and 6D). Several of these GO terms, including post-embryonic development (BP1), molecular transport/localization (BP3), cell cycle (BP5), molecular catabolism/metabolism (BP7/8) and microtubule binding (MF1), were also reported in a previous study analyzing vegetally enriched transcripts in the *Xenopus* oocyte [35]. Notably, we found 45 germ granule positive/vegetal tier-enriched transcripts under the reproductive process GO term (BP1), and four of these, including *nanos*/85485, *sycp3*/117694, *tbx2*/124781 and *ubc*/113894, were confirmed to be localized to the amphioxus germ granule (Fig 4). In addition, several kinesin and dynein-associated transcripts, including *dctn5*/85198, *kif1c*/86166, *dnal4*/119023, *kif19*/133981 and *kif15*/201006 are present under GO terms single-organism cellular localization (BP3) and microtubule binding (MF1). Thus, GO enrichment analysis suggested that transcripts related to mRNA metabolism and dynein/kinesin-mediated intracellular transportation may play roles in controlling the asymmetric distribution of maternal transcripts. Additionally, consistent with our previous

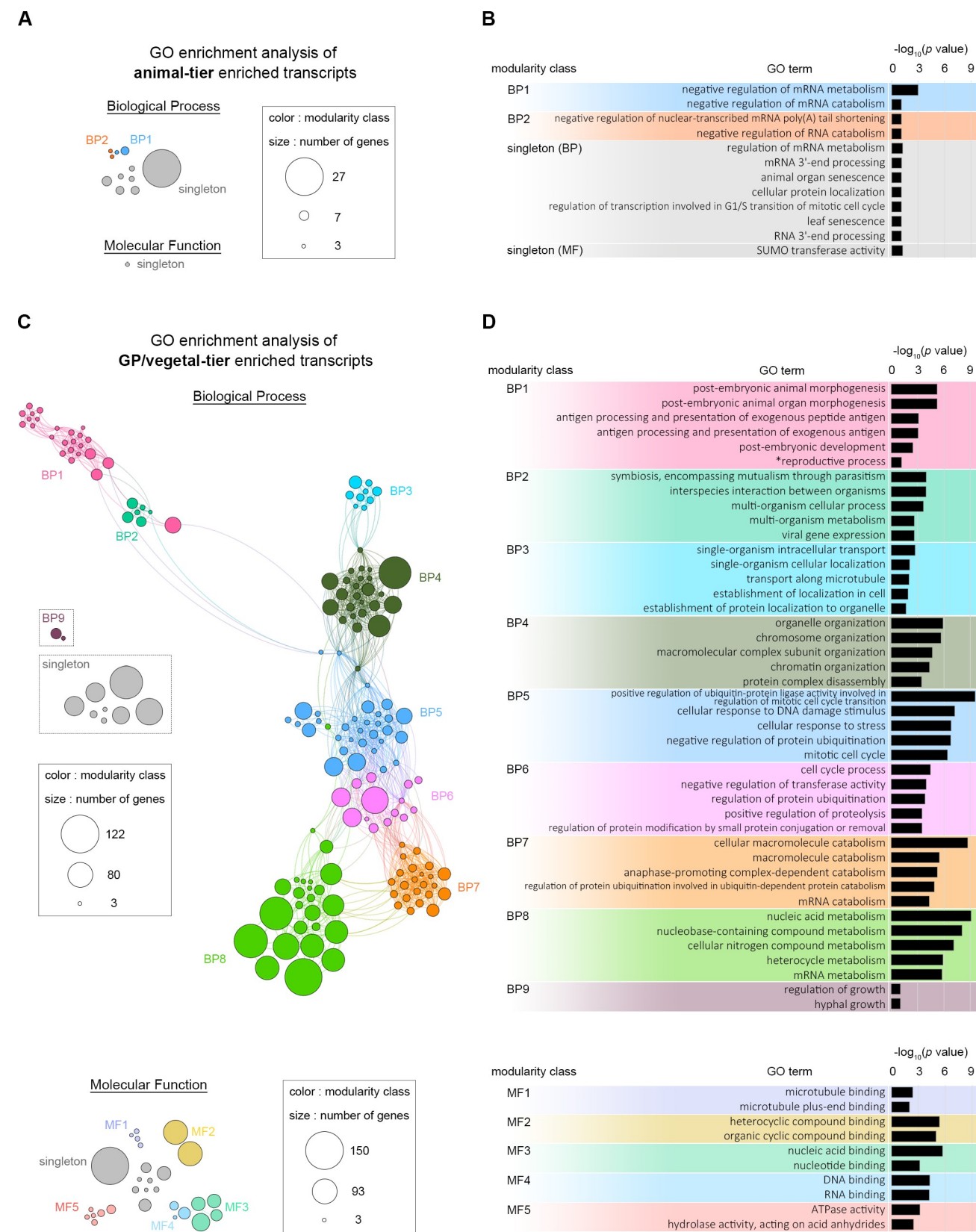

**Fig 6. Gene ontology (GO) enrichment analyses of the DETs.** (A) Results of the GO enrichment network analysis (adjusted *p* value < 0.1) of animal tier-enriched transcripts. Individual node of the network denotes a specific enriched GO term, including biological process (BP) and molecular function (MF). The GO terms are clustered and divided into different modules that are labeled with different colors. Unclassified GO terms are labeled as singletons. Sizes of the circles indicate numbers of genes in each GO term. (B) Descriptions of the most enriched GO terms within each module for the animal tier-enriched transcripts. The bars indicate -log10 adjusted *p* values for the corresponding GO terms. (C) Results of the GO enrichment network analysis (adjusted *p* value < 0.1) of germ granule-positive and/or vegetal tier-enriched transcripts. Individual node of network denotes a specific enriched GO term, including biological process (upper panel) and molecular function (lower panel). Lines connecting different nodes represent their semantic similarity. (D) Descriptions of GO terms within each module with the -log10 adjusted *p* value. Five GO terms with the smallest adjusted *p* values from different modules are presented. One manually selected GO term is indicated (*).

observations, many of the germ granule positive/vegetal tier-enriched transcripts are related to reproductive process and germ cell development.

## Overexpression of amphioxus *zfp665* and *soxB1b* dorsalizes zebrafish embryos

Asymmetric distribution of transcription factor-encoding transcripts in early embryos often imply the involvement of such transcripts in specification of body axes and germ layers. We thus set out to examine the molecular functions of four newly identified amphioxus genes with asymmetrically localized transcripts. Because gene-specific functional experiments are technically challenging in amphioxus embryos, we used zebrafish embryos as a surrogate *in vivo* model, with which amphioxus shares similar axial patterning mechanisms [21,36]. Synthesized mRNA of amphioxus *zfp665*/198050, *soxB1b*/130889, *tbx2*/124781 and *foxn1/4b*/247432 were injected separately into zebrafish zygotes, and the resulting phenotypes were assessed at 2 to 3 days post-fertilization. In amphioxus embryos, *zfp665*/198050 was enriched in animal/anterior blastomeres (Fig 5), which later contribute to the dorsal/anterior region. Consistently, we found that embryos injected with amphioxus *zfp665* mRNA displayed a dose-dependent dorsalized/anteriorized phenotype compared to the embryos injected with *gfp* mRNA (control) (Fig 7A and 7B). In some extreme cases, the injected embryos exhibited incomplete dual axes (arrows in Fig 7A). We further examined expression of the genes that mark dorsally derived tissues, including *pax2.1* (marking midbrain-hindbrain boundary), *krox20* (labeling rhombomeres 3 and 5), *myoD* (dorsal mesoderm), and *six3a* (forebrain) [37–40]. We observed clear lateral expansion of the midbrain-hindbrain boundary and rhombomeres 3 and 5 in the *zfp655* mRNA-injected zebrafish embryos at the 10 somite stage (Fig 7C). Quantitative analysis showed that the length of rhombomere 5 was significantly increased when *zfp655* was overexpressed (*t*-test, *p* < 0.001, Fig 7D). These changes in the gene expression patterns confirmed the observed dorsalization phenotype.

Based on our RNA-seq data, *soxB1b* is enriched in the germ granule-positive blastomere (Fig 3); in addition, *in situ* hybridization experiments indicate that it is enriched toward the animal pole [31]. Overexpression of amphioxus *soxB1b* in zebrafish embryos also caused a dorsalized phenotype (Fig 7E and 7F), and a higher dose of *soxB1b* resulted in the loss of one or both eyes in 41% of the surviving embryos (n = 37; S7 Fig). Gene expression analysis at the 10 somite stage showed that the length of rhombomere 5 was significantly increased in *soxB1b* mRNA-injected zebrafish embryos (Fig 7G and 7H, *t*-test, *p* < 0.001). These findings are similar to previous studies showing that the *soxB* family is required for dorsoventral patterning [41,42] and overexpression of *sox3* leads to eye defects in zebrafish embryos [42]. In contrast, overexpression of amphioxus vegetal/germ granule-enriched *tbx2* or *foxn1/4b* did not cause detectable axial defects in zebrafish embryos, even when a much larger amount of mRNA was injected (200 pg, Fig 7I). Overall, our results showed that overexpression of the amphioxus *zfp665* and *soxB1b* is able to dorsalize zebrafish embryos, consistent with their expected functions based on their transcript distribution patterns in the amphioxus embryo. On the other

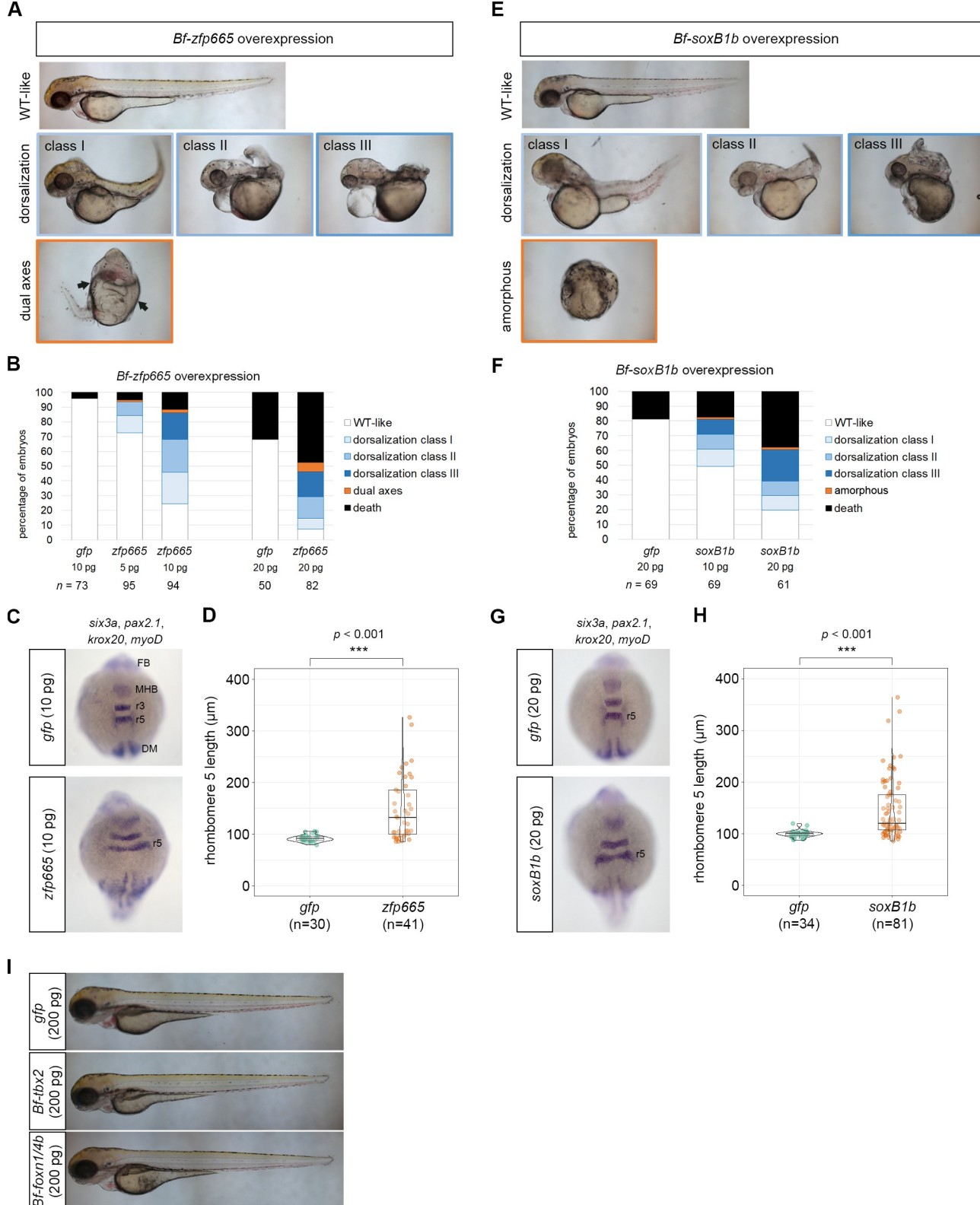

**Fig 7. Functional validation of four asymmetrically localized transcripts encoding transcription factors.** (A) Phenotypes of the 48 hpf zebrafish embryos injected with amphioxus *zfp665* mRNA. (B) Percentages of 48 hpf zebrafish embryos with the indicated phenotypes following injection of *gfp* (control) or amphioxus *zfp665* mRNA. (C) WMISH of *pax2.1*, *krox20*, *myoD* and *six3a* in gfp (control) or amphioxus *zfp665* mRNA-injected zebrafish

embryos at the 10 somite stage. (D) Statistical analysis of the lengths of rhombomeres 5 in *gfp* and *zfp665* mRNA injected zebrafish embryos at the 10 somite stage. (E) Phenotype of the 48 hpf zebrafish embryos injected with amphioxus *soxB1b* mRNA. (F) Percentages of 48 hpf zebrafish embryos with the indicated phenotypes after injection of *gfp* (control) or amphioxus *soxB1b* mRNA. (G) WMISH of *pax2.1*, *krox20*, *myoD* and *six3a* in gfp (control) or amphioxus *soxB1b* mRNA-injected zebrafish embryos at the 10 somite stage. (H) Statistical analysis of the lengths of rhombomeres 5 in *gfp* and *soxB1b* mRNA-injected zebrafish embryos at the 10 somite stage. (I) Embryos injected with amphioxus *tbx2* or *foxn1/4b* mRNA did not show any detectable axial defects compared to control embryos at 72 hpf. DM, dorsal mesoderm; FB, forebrain; MHB, midbrain-hindbrain boundary; r3, rhombomere 3; r5, rhombomere 5. Statistical significance was determined by Student's *t*-test. ***p < 0.001. Underlying data are available in S1 Data.

hand, ectopic amphioxus *tbx2* and *foxn1/4b* were unable to elicit axial defects in zebrafish embryos, suggesting that they are not involved in axial patterning, or their functions have diverged between these two species.

## Amphioxus zfp665 represses zebrafish ventral gene expression

To further understand how amphioxus *zfp665* disrupts axial patterning in zebrafish embryos, we examined the expression of several ventral and dorsal marker genes during gastrulation (Fig 8). In most zebrafish embryos with exogenous amphioxus *zfp665*, the expression domain of the ventral gene *eve1* was significantly reduced (quantified by measuring the angles of the signals observed from the animal pole; *t*-test, *p* < 0.001, Fig 8A), while expression of other ventral genes, *vox*, *vent* and *bmp2b*, and two dorsal marker genes, *goosecoid* (*gsc*) and *chordin* (*chd*), were not significantly affected (Fig 8B–8F). These results suggest that *zfp665* disrupts dorsoventral patterning via repression of *eve1* expression. Consistently, it has been shown that depletion of *eve1* and/or *vox*/*vent* in zebrafish results in a dorsalized phenotype [43,44].

We next wanted to examine whether *zfp665* functions as a transcriptional repressor or activator. Therefore, we generated an obligate repressor by fusing the zinc finger domain (DNA binding domain) of zfp665 to the engrailed repressor domain (*En-zfp665ΔC*), and we generated an activator by fusing the zfp665 DNA binding domain with the VP16 activation domain (*VP16-zfp665ΔC*) (Fig 9A and 9B). Similar to the effect of full-length *zfp665*, *En-zfp665ΔC* mRNA dorsalized zebrafish embryos (33%, n = 55) (Fig 9C and 9D) and significantly extended the length of rhombomere 5 (Fig 9E and 9F). Conversely, *VP16-zfp665ΔC* mRNA resulted in a slightly ventralized phenotype, with expansion of the ventral tissues (21%, n = 81); on the other hand, overexpression of the zinc finger domain alone (*zfp665ΔC*) did not cause any detectable axial defects (Fig 9C and 9D). Consistently, the length of rhombomere 5 was not affected by overexpression of *VP16-zfp665ΔC* or *zfp665ΔC* mRNA (Fig 9E and 9F). Together, these data suggest that the ectopically expressed amphioxus *zfp665* functions as a transcriptional repressor, possibly by repressing *eve1* expression, to interfere zebrafish dorsoventral patterning.

## Differential expression analysis between the 2- and 8-cell stages

In addition to identifying DETs in isolated blastomeres at the same stage, our RNA-seq datasets allowed us to examine early zygotic gene activation from the 2- to the 8-cell stage in the amphioxus embryo. We combined RNA-seq reads from isolated blastomeres of the same stage and performed differential expression analysis between the two stages (S8A and S8B Fig). We found 1406 genes with significantly higher transcript levels at the 8-cell stage compared to those at the 2-cell stage, using both DESeq2 and edgeR methods (S8C Fig). These genes may be activated between the 2- and 8-cell stages, and 19 of the upregulated transcripts were differentially enriched either in the animal (14 transcripts) or vegetal tier (5 transcripts) at the 8-cell stage (S8D Fig). Intriguingly, one of the five vegetal DETs that showed higher levels at the 8-cell stage was *kif4*, which had transcripts associated with the granule at the 8- but not the 2-cell stage (Fig 4Ca). Therefore, it is possible that the zygotic activation of *kif4* also contributes

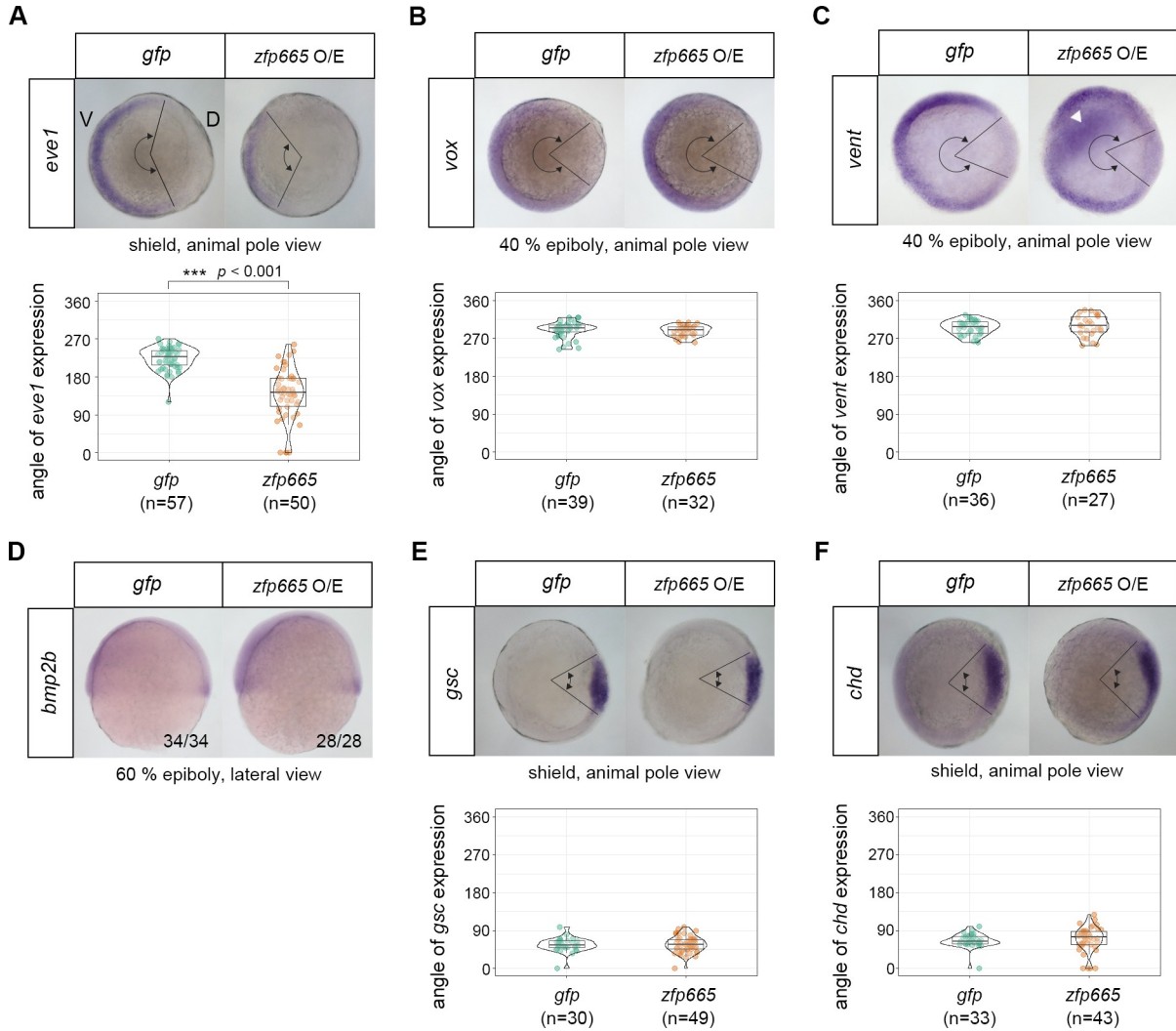

**Fig 8. Expression domain of the ventral gene, *eve1*, was significantly reduced in *zfp665* overexpressed zebrafish embryos.** (A-F) WMISH of zebrafish dorsal-ventral genes *eve1* (A), *vox* (B), *vent* (C), *bmp2b* (D), *gsc* (E) and *chd* (F), in *gfp* (control) or amphioxus *zfp665* mRNA-injected zebrafish embryos. The stages and views of the embryos are indicated below the images. Statistical analyses of *eve1*, *vox*, *vent*, *gsc* and *chd* expression domains in embryos injected with *gfp* and *zfp665* mRNA. The expression domains were quantified by measuring the angles of the signals observed from the animal pole. Ectopic expression of *vent* (arrowhead) was observed in some *zfp665*-overexpressing embryos (44%, n = 27) (C). The ratio of embryos exhibiting the displayed *bmp2b* expression patterns is indicated in the bottom right-hand corner (D). V, ventral side; D, dorsal side. Statistical significance was determined by Student's *t*-test, ***p < 0.001. Underlying data are available in S1 Data.

to the compact signal observed at the 8-cell stage. GO enrichment analysis revealed that transcripts showing higher levels at the 8-cell stage were mainly associated with regulation of the cell cycle (S8E Fig), suggesting that the majority of early activated genes are involved in cell cycle control.

On the other hand, our differential expression analysis also revealed 1604 transcripts that showed lower levels at the 8-cell stage in comparison to the 2-cell stage (S8F Fig). These transcripts may be degraded between the 2- and 8-cell stage, and 55 were differentially enriched either in the animal or vegetal tier at the 8-cell stage (S8G Fig). Among these 55 DETs, the majority (51 transcripts) were enriched in the vegetal tier (S8G Fig), suggesting that vegetal DETs are preferentially degraded between the 2- and 8-cell stages. For example, *sycp3* and

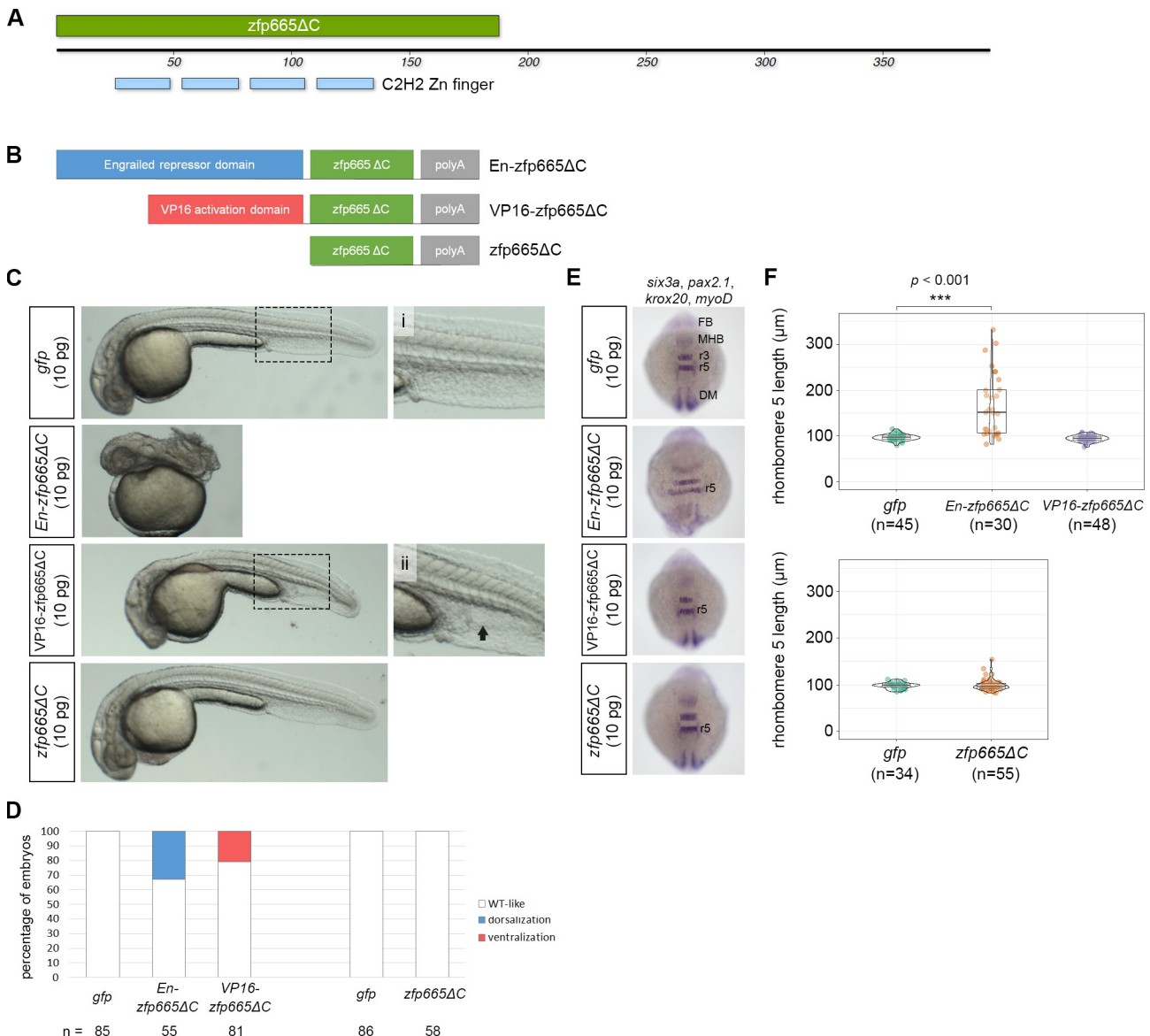

**Fig 9. Functional validation of *zfp665* as a transcriptional repressor in zebrafish embryos.** (A) A schematic representation of the protein domain organization of amphioxus zfp665, showing four consecutive C2H2-type zinc finger domains at the N-terminus. zfp665ΔC denotes the zinc finger region of zfp665 without its C-terminus. (B) A schematic representation of three different zfp665 fusion proteins, including engrailed repressor domain-zfp665ΔC fusion protein (En-zfp665ΔC), VP16 activation domain-zfp665ΔC (VP16-zfp665ΔC) fusion protein and the deletion of C-terminus of zfp665 (zfp665ΔC). (C) Phenotype of the zebrafish embryos injected with one of the three *in vitro* synthesized mRNA encoding the zfp665 fusion proteins at 24 hpf. (i) and (ii) are high-magnification views of the corresponding boxed areas. The black arrow indicates expanded ventral tissue. (D) Percentages of 24 hpf zebrafish embryos with the indicated phenotypes after injection of *gfp* (control) or one of the three *in vitro* synthesized amphioxus mRNAs encoding zfp665 fusion proteins. Only live larvae were scored. (E) WMISH of *pax2.1*, *krox20*, *myoD* and *six3a* in *gfp* (control) or indicated amphioxus mRNA-injected zebrafish embryos at the 10 somite stage. (F) Statistical analysis of the lengths of rhombomere 5 at the 10 somite stage in zebrafish embryos injected with *gfp* or one of the three forms of *zfp665* mRNAs. DM, dorsal mesoderm; FB, forebrain; MHB, midbrain-hindbrain boundary; r3, rhombomere 3; r5, rhombomere 5. Statistical significance was determined by Welch's one-way ANOVA with Games-Howell post hoc test plus Bonferroni correction when the number of groups was greater than 2 or Student's *t*-test when the number of groups was 2. ***p < 0.001. Underlying data are available in S1 Data.

*foxn1/4* were among the 51 transcripts, and WMISH confirmed that the transcript levels were decreased in the germ granule-negative blastomeres at the 8-cell stage (Fig 4Cg and 4Ci). GO enrichment analysis supported the idea that vegetal DETs may be preferentially degraded, as

transcripts showing lower levels at the 8-cell stage were largely involved in RNA metabolism (S8H Fig).

## Discussion

In this study, we systematically surveyed transcript distributions in paired samples of amphioxus blastomeres at the early cleavage stages using a PCR-based scRNA-seq approach. We identified 111 transcripts that were differentially enriched in the germ granule-positive or -negative blastomere at the 2-cell stage, and 391 transcripts that were differentially enriched in the animal or vegetal tier at the 8-cell stage. We found that these maternal transcripts can be categorized into two major groups: (1) vegetal tier/germ granule-enriched and (2) animal tier/anterior-enriched; a subset of these transcripts were validated by WMISH. In sum, our study provides a global transcriptomic blueprint for early cleavage stage amphioxus embryos.

ScRNA-seq profiling is a powerful approach for studying animal development, nevertheless, we needed to overcome several limitations and difficulties when applying this technology to identify DETs in early amphioxus embryos (see Methods). First, because the initial cell divisions are symmetric in amphioxus embryos, it was impossible to assign cell identities by morphology. To solve this problem, we used qPCR to identify cell types and removed ambiguous samples before sequencing. Therefore, we used the post-barcoded Smart-seq2 scRNA-seq system, rather than the CEL-seq or droplet-based pre-barcoded systems, in order to perform qPCR prior to barcoding. Using this approach, we concluded that our qPCR-based identification process was consistent with the RNA-seq data. Second, we noticed strong batch effects and individual differences when analyzing amphioxus transcripts. A similar problem has also been reported for RNA-seq analyses of the ctenophore *Mnemiopsis leidyi* and the sea urchin *Strongylocentrotus purpuratus* [45,46]. A paired sample test with multifactor design was effective in reducing individual differences, so that DETs could be reliably identified. Third, we noticed that fold differences of the DETs identified in our study never exceeded 3, a difference that is lower than those reported in sea urchin (maximum 4-fold between the micromeres and the animal pairs) [45] and ascidian (maximum 7.78-fold between animal and vegetal blastomeres) [7]. The highest fold differences we identified were 2.42 for germ granule-enriched transcripts at the 2-cell, 2.51 for animal tier/anterior-enriched transcripts at the 8-cell stage, and 1.98 for vegetal tier-enriched transcripts at the 8-cell stage (based on DESeq2). These small fold differences could be a result of bias in the PCR-mediated cDNA amplification. Additionally, some germ granule-enriched transcripts show weak and diffuse distributions in the germ granule-negative cell along with concentrated signals in the germ granule; this diffuse expression may be the cause of low fold differences [19,20]. Despite these limitations and difficulties, we have identified hundreds of animal tier/anterior-enriched and vegetal tier/germ granule-enriched transcripts in early amphioxus embryos. It should be noted that because distributions of transcripts may not be homogenous in each of the blastomeres within the animal and vegetal tiers of the 8-cell stage embryo, validation by WMISH is essential to reveal the distributions of DETs identified by our RNA-seq analyses. For example, *foxn1/4b*, one of the DETs in the germ granule and vegetal tier, exhibited an additional asymmetrical distribution pattern among the vegetal blastomeres (Fig 4Ci). On the other hand, although *dag1* was identified to be enriched in the animal tier, WMISH revealed that in addition to its uniform distribution in the animal blastomeres, its transcripts also aggregated in one vegetal blastomere (Fig 5Cj). These complex distribution patterns suggest that the transcripts may play multiple roles in amphioxus embryogenesis, and further characterizations of the identified DETs will be necessary to understand their functions.

Previous studies have shown that transcripts of amphioxus germline determinants aggregate into a condensed granule soon after fertilization [19,20,32]. Here, based on our RNA-seq

data, we found additional 100+ transcripts that are enriched in the germ granule-positive blastomere, and at least 13 of these have been confirmed to be colocalized within the germ granule. GO analysis revealed that transcripts of dynein and kinesin genes are enriched in the germ granule-positive and/or vegetal tier blastomeres (Fig 6). This finding is consistent with discoveries made in *Drosophila* and zebrafish embryos where dynein and kinesin motor proteins transport germ plasm transcripts along microtubules [4,47,48], suggesting that these motor proteins may play similar roles during assembly of the amphioxus germ granule. In addition to active transport, asymmetric localization of mRNAs can also be achieved via differential activation of RNA degradation. In *Drosophila*, *Hsp83* transcripts are initially ubiquitously distributed throughout fertilized eggs, but 2–3 h after fertilization, the transcripts are degraded in the anterior region, restricting them to the posterior germ plasm. This degradation is triggered by Smaug protein binding to hairpin structures in the 3' UTR of *Hsp83* and the subsequent recruitment of CCR4/POP2/NOT deadenylase complex [49–51]. In *C. elegans*, the decapping activators LSM-1 and LSM-3 and CCR4-NOT deadenylase complex are also required for the spatial and temporal degradation of maternal transcripts in somatic P-bodies [52,53]. Interestingly, we found that one of the vegetal tier-enriched GO terms "mRNA catabolism" contains one CCR4/POP2/NOT deadenylase complex subunit component, cnot3, and two decapping activators, lsm3 and lsm4, suggesting that RNA degradation may also affect transcript distribution in amphioxus early embryos. We also found that a major function of the animal tier-enriched transcripts is "negative regulation of mRNA metabolism/catabolism", including a poly-A binding protein—pabp1, suggesting that mRNA protection mechanisms also regulate asymmetrical distribution of certain mRNAs. Based on the GO enrichment analyses, we hypothesize that localizations of animal tier-enriched and vegetal tier/germ plasm-enriched transcripts are determined by different cellular mechanisms. The asymmetrical localization of the vegetal tier/germ granule-enriched transcripts might heavily rely on dynein/kinesin transport, while increased mRNA stability in the animal-tier and differential degradation in the vegetal-tier may be a major driving force for animal tier-enriched transcripts. Moreover, we found evidence of early zygotic gene activation from 2- to 8-cell stages (S8 Fig), which is earlier than previously reported activation at the early blastula (64- to 128-cell) stage [18]. Therefore, zygotic gene expression may also contribute to the asymmetric distributions of transcripts in the early amphioxus embryos. It should be noted that our analyses were based on measurements of polyadenylated transcripts and did not include transcripts without polyadenylation. It was previously shown that specific maternal mRNAs in *Xenopus* are polyadenylated or deadenylated after fertilization [54]. RNA-seq analyses in the marine annelid *Platynereis dumerilii* also suggest that polyadenylation may partly contribute to changes in developmental expression profiles shortly after fertilization [55]. Thus, it is possible that some of the "early activated genes" we identified are actually maternally loaded transcripts that are polyadenylated between the 2- to the 8-cell stage. Additional RNA-seq experiments using mRNAs captured by different methods (e.g., [56]) would be required to further address this issue.

We found that several animal tier/anterior-enriched transcripts displayed a gradient distribution from the animal/anterior blastomeres to the vegetal/posterior blastomeres (Fig 5C and 5E). A similar gradient pattern has also been described for *nodal* transcripts in another amphioxus species, *Branchiostoma japonicum* [17]. Using *foxq2* as an anterior marker, previous studies have indicated that the anterior tip of the amphioxus gastrula is offset dorsally by about 20 degrees from the animal pole [57,58], implying a close relationship between the anteriorly enriched transcripts and the establishment of the dorsal structures. Indeed, we found that overexpression of one such transcript, *zfp665*, was able to dorsalize/anteriorize zebrafish embryos, possibly via downregulating *eve1* expression on the ventral side. Zfp665 belongs to the zinc finger protein family, and members of this family possess diverse domain structures

that obscure the phylogenetic affiliations in most organisms. The amphioxus zfp665 contains four sequential C2H2 zinc finger DNA binding domains at its N-terminus (Fig 9A), and we were unable to identify a clear orthologue in zebrafish or other species that we have searched. Nevertheless, several zinc finger domain-containing proteins have been found to regulate axial patterning in *Drosophila* [59], zebrafish [60] and *Xenopus* [61]. We hypothesize that the over-expressed amphioxus zfp665 may behave like some unidentified endogenous C2H2 zinc finger protein to control zebrafish *eve1* expression. In amphioxus embryos, expression of an *eve1*-related gene, *evx*, is also restricted to the posterior-ventral side [21]. Homologs of *evx* in vertebrates have been shown to be involved in patterning the anterior-posterior axis [62,63]. Further studies are needed to confirm whether zfp665 represses *evx* expression on the anterior-dorsal side during amphioxus embryogenesis. Another transcript that showed enrichment in the amphioxus animal tier/anterior blastomeres and caused dorsalization in zebrafish embryos encodes transcription factor soxB1b, which is one of the soxB1 paralogs in amphioxus [64]. It has been reported that members of the zebrafish soxB1 family (sox1, sox2 and sox3) are initially expressed in the ectodermal cells toward the animal pole and later become more restricted to the future neuroectoderm [65], and these factors are required for dorsoventral patterning [41,42]. Thus, it is plausible that overexpression of amphioxus *soxB1b* mimicked the effects of the endogenous zebrafish soxB family genes. Further experimental studies on soxB1b in amphioxus embryos will be required to confirm its function in dorsoventral patterning.

In conclusion, by using the PCR-based scRNA-seq approach, we have successfully generated a spatial transcriptomic map for early amphioxus embryos. These results clearly demonstrate the mosaic property of the early amphioxus embryo, despite the fact that its embryogenesis is generally considered to be highly regulative. More importantly, this dataset can serve as a conceptual platform for guiding future studies on specific molecular players in early amphioxus development. The GO enrichment analyses provide a global view of potential functions for the DETs. Because amphioxus represents an early-branching group within the chordate animals, further comparative studies may provide important insights into the evolution of developmental mechanisms for chordate body plans.

## Methods

### Ethics statement

All animal procedures were approved by the Academia Sinica Institutional Animal Care & Use Committee (AS IACUC) (Protocol ID: 15-12-918; BSF0412-000002656). All methods were performed in accordance with the approved guideline.

### Adult animals and embryos

Adult amphioxus *Branchiostome floridae* and zebrafish were raised under standard conditions, as previously described [66,67]. Amphioxus embryos were obtained from spontaneous spawning events and maintained at 25˚C. Zebrafish embryos were maintained at 28.5˚C, and the morphological criteria were defined as previously described [68].

### Blastomere separation and reverse transcription

After removing the fertilization envelope, the left/right blastomeres from the 2-cell stage and animal/vegetal tier blastomeres from the 8-cell stage amphioxus embryos were separated manually using hair or hand-pulled glass pipettes. Subsequently, RNA-seq was conducted on the separated blastomeres according to the Smart-seq2 method [23,24]. Each separated blastomere

(~0.3 μl) was pipetted into a 0.2 ml PCR tube containing 2 μl of lysis buffer (0.2% Triton X-100 and 2 U/μl RNase inhibitor, Promega) and stored at −80˚C or processed immediately. After adding 1 μl of 10 mM dNTP (Kapa Biosystems) and 1 μl of 10 μM Oligo-dT30VN primer (5'-AAGCAGTGGTATCAACGCAGAGTACT30VN-3', IDT), samples were incubated at 72˚C for 3 min, then immediately spun down and placed on ice. Subsequently, the reverse transcription mixtures were assembled by mixing the samples with a 6.9 μl reverse transcription master mix containing 2 μl of 5x SuperScriptII first-strand buffer (Invitrogen), 0.25 μl of 100 mM DTT (Invitrogen), 2 μl of 5 M betaine (Sigma), 0.9 μl of 100 mM MgCl$_2$, 1 μl of 10 μM template-switching oligonucleotide (TSO, 5'-AAGCAGTGGTATCAACGCAGAGTACATrGrG +G-3', Exiqon), 0.25 μl of 40 U/μl RNase-inhibitor (Promega) and 0.5 μl of 200 U/μl SuperScriptII reverse transcriptase (Invitrogen). The reverse transcription reaction was carried out using the following program: 42˚C for 90 min, 10 cycles of [50˚C for 2 min plus 42˚C for 2 min], and 70˚C for 15 min.

## PCR amplification

After reverse transcription, the PCR master mix containing 12.5 μl of 2X KAPA HiFi HotStart ReadyMix (Kapa Biosystems), 1 μl of 10 μM IS PCR primers (5'-AAGCAGTGGTATCAACG CAGAGT-3', IDT) and 0.3 μl of nuclease-free water (Ambion) was added into each sample. Subsequently, PCR was conducted using the following program: 98˚C for 3 min, 18 cycles of [98˚C for 15 s, 67˚C for 20 s and 72˚C for 6 min] and 72˚C for 5 min. cDNA purification was carried out using the Ampure XP beads (Beckman), following the standard protocol with a 1:1 ratio of cDNA solution and beads. The beads were then resuspended in 17.5 μl EB (Qiagen) and immobilized on a magnetic stand, and 15 μl of the supernatant was taken for analysis.

## Quantitative real-time PCR (qPCR)

qPCR was conducted by using 2X SYBR Green PCR Master Mix (Roche) with a Roche Light Cycler 480 II thermal cycler. A 10X dilution of the pre-amplified cDNA was used with the following program: 95˚C for 5 min for pre-incubation, 40 cycles of [95˚C for 10 s, 60˚C for 10 s and 72˚C for 10 s] for amplification and one cycle of [95˚C for 5 s, 65˚C for 1 min and continuous heating to 97˚C] for melting curve, followed by cooling at 40˚C for 30 s. The following genes were assayed using the indicated primer pairs: *vasa* (5'-CCAATACCATGCCCAAGA CT-3' and 5'-ACAAACCAAGTGCCTTCACC-3'); *nanos* (5'-CCCTGTCCTTATGGCCTA CA-3' and 5'-GTTGGGTAGCTGGTTGGTGT-3'); *piwil1* (5'-CCACTTCTTTGCTCAGGT CTAC-3' and 5'-GATACGTCCGGAGTTGATGTTT-3'); *tdrd*/118019 (5'-ATCCAGCCAG CAGTCCTCATCA-3' and 5'-GACTGCTGCTCCACCATCTCTG-3'); *gene model* 130738 (5'-CCACCAGAGCATAGCCCTTGAG-3' and 5'-GGCTGGATCCTGTTTGTGACTG-3'); *tdrd*/176882 (5'-GGCGAGACACGAGCGAAAAA-3' and 5'-CGTCCGAAAACTGTAGCG TCAA-3'); *gene model* 126022 (5'-AGCCTTGTCCATCTTCATAGT-3' and 5'-AGTGTGCC AACCGTCTGT-3'); *gene model* 68495 (5'-ACGCCGACGAGTGGTTTGA-3' and 5'-TTGC CTTGGTTACGGTGACTGT-3'); *gapdh* (5'-AAATGGGCGGAAGCAGGTG-3' and 5'-AC ATCGGCGCATCAGCAGAG-3'); *β-actin* (5'-ACCCCGTGCTGCTGACTGAG-3' and 5'-CA TGGCTGGGCTGTTGAAGG-3').

## Quality assurance on unamplified cDNA

Concentrations of unamplified cDNA samples were measured using a Qubit fluorometer (Thermo Fisher Scientific). The concentrations ranged from 10 to 60 ng/μl. The size distribution was verified by a Bioanalyzer 2100 with a High Sensitivity DNA Analysis Kit (Agilent). The ideal size distribution is a bell curve with a peak at ~1.5–2 kb. Based on the qPCR data

showing expected patterns of primordial germ cell markers and ideal size distributions of cDNAs, six sets of 2-cell stage embryos (12 samples) and six sets of 8-cell stage embryos (12 samples) were selected for library preparation.

## Library preparation

For each sample, ~40 ng of the unamplified cDNA was subjected to library preparation by using a Nextera DNA Library Prep Kit (Illumina) with the manufacturer's protocol, including the tagmentation reaction [55˚C for 3 min] and indexing PCR for engineering adaptors and sample-specific barcodes by using the following program: 95˚C for 30 s, 5 cycles of [95˚C for 10 s, 55˚C for 30 s and 72˚C for 30 s] and 72˚C for 5 min. The amplified cDNA libraries were then purified with Ampure XP beads in a 0.6:1 ratio of cDNA to beads. Subsequently, the Qubit fluorometer and Bioanalyzer were used to measure concentrations and verify size distributions of the 24 cDNA libraries.

## RNA-seq analysis

The cDNA libraries were subjected to multiplexed sequencing by paired-end 2x181 nt format on a HiSeq 2500 sequencer (Illumina) in 3 separate lanes. The average read depth was 25,452,693 paired reads per sample. Read quality trimming was applied to remove the Illumina Nextera indexed adaptor and low-quality bases using Trimmomatic (version 0.35) [69]. The trimmed reads were then mapped to *B. floridae* reference sequences including genome assembly v2.0 (28,666 gene models) and mitochondria DNA (Branchiostoma_floridae_v2.0.assembly.fasta.gz downloaded from JGI, and http://hgdownload.cse.ucsc.edu/goldenPath/braFlo1/chromosomes/chrM.fa.gz, downloaded from UCSC) using Stampy software (version 1.0.28) with a 0.05 of substitution per site (—substitutionrate = 0.05) [70]. The read alignment file combined with the corresponding gene model file (Bfloridae_v1.0_FilteredModelsMapped-ToAssemblyv2.0.gff.gz), which had been downloaded from the JGI database, was subjected to raw counts estimation using HTSeq software (version 0.6.1) [71] and resulted in an average of 5,840,180 counts for each. Saturation analysis was conducted by using NOISeq (version 2.16.0) [72]. The DESeq2 R package (version 1.12.4) was then applied to conduct a count-based quality check and depict relationships between samples, including outlier identification by Cook's distance, dispersion estimation, principal component analysis and hierarchical clustering by Euclidean distances [73]. For two principal component analyses (Fig 1E and 1I), the limma R package (version 3.34.9) [74] was used for normalization based on embryo identification.

## Differentially enriched transcripts and gene ontology enrichment analyses

The raw count-based gene expression matrix was analyzed with DESeq2 (version 1.12.4) [73] and edgeR (version 3.14.0) [75]. A paired sample test with multifactor design was used for both DESeq2 and edgeR; the first factor was set as embryo identification and the last factor was set as cell type. The adjusted *p*-value (FDR) threshold was set to a default value of 0.1. Venn diagrams were generated using Venn Diagram Plotter software (https://omics.pnl.gov/). The intersections of DETs identified by DESeq2 and edgeR were subjected to gene ontology (GO) analyses using the online tools, GOEAST (http://omicslab.genetics.ac.cn/GOEAST/tools.php) [76]. A customized amphioxus GO reference [25] was used, and the adjusted *p*-value (FDR) threshold was set to a default value of 0.1. The REVIGO algorithm (http://revigo.irb.hr/) [77] was then used to remove redundant GO terms. Finally, the enriched GO terms were clustered and their network was visualized based on the semantics by Gephi (version 0.9.1, https://gephi.org/). The interactive websites for searching DETs were generated based on

the Highcharts javascript library (https://www.highcharts.com/) and scripts were written in-house.

## Whole-mount *in situ* hybridization

Amphioxus embryos were fixed in 4% paraformaldehyde for 1 h at room temperature and then subjected to WMISH using digoxigenin (DIG)-labeled antisense RNA probes and alkaline phosphatase-conjugated anti-DIG antibody, as previously described [19]. Various templates derived from the EST clones constructed in the pDONR-222 vector [27] were PCR amplified using the primer pairs: 5'-ATTTAGGTGACACTATAGAAGACGGCCAGTCT TAAGCTC-3' and 5'-TAATACGACTCACTATAGGGAGGGGATATCAGCTGGATG-3'. Antisense RNA probes were synthesized *in vitro* using T7 RNA polymerase (Promega). Double-fluorescent *in situ* hybridization was performed as described previously [19]. DIG-labeled antisense RNA probes for *gene models* 131234, 275420, 127160 and 124781 and a fluorescein-labeled antisense RNA probe for *nanos* were synthesized. Anti-DIG-POD and anti-Fluorescein-POD antibodies (Roche) were used to detect the RNA probes, followed by fluorescence detection by using a TSA Plus Cyanine 3 and Fluorescein Evaluation Kit (PerkinElmer). Zebrafish embryos were fixed in 4% paraformaldehyde overnight at 4˚C, then subjected to WMISH using DIG-labeled antisense RNA probes and alkaline phosphatase-conjugated anti-digoxigenin antibody, as previously described [78]. Various templates derived from pGEMT, pGEMT-Easy, or pBluescript SK(+) vectors were linearized by restriction enzyme digestion or PCR amplification, and the following antisense RNA probes were generated (restriction site or PCR primer pairs and promoter in parentheses): *bmp2b* (Eco RI/T3, Promega), *vox* (Bam HI/T3), *vent* (Bam HI/T3), *eve1* (M13F and M13R primer pairs/T7), *gsc* (T7 and SP6 primer pairs/T7) and *chd* (T7 and SP6 primer pairs/SP6, Promega). Images of amphioxus and zebrafish embryos were taken by using a Zeiss AxioCam MRC camera mounted on a Zeiss Imager A2 microscope or a Zeiss confocal system (Zeiss LSM 880 with airyscan). Photoshop CC (Adobe) was used to assemble all figures and minimally adjust the brightness of images.

## Plasmid constructions

Full-length coding sequences of amphioxus *zfp665*, *soxB1b*, *tbx2* and *foxn1/4b* and *zfp665ΔC* were amplified by PCR using appropriate primer pairs: zfp665 (5'-GAGGGATCCATGCCG GCTGCTGTACGGAG-3' and 5'-GGAGTGAATTCTTATAAGTATCTCTCCAGCTCGA G-3'); soxB1b(5'-CCGGGATCCATGATGATGATGCACCCCCCAG-3' and 5'-GACAATC GATATGACTGTTCTAGCTTGCCGAC-3'); tbx2 (5'-CTGGATCGATTTATAATACCATT CCAGGAGGCA-3' and 5'-GGAGTGAATTCTTATAAGTATCTCTCCAGCTCGAG-3'); foxn1/4b (5'-AGGGGATCCATGTACGCGAACTCGCTGCCGG-3' and 5'-TGCAATCGAT TTAGTACATGGTAAGCTGTGCTC-3') and zfp665ΔC (5'-CGAGCTCGAGATGCCGGCT GCTGTACGGAG-3' and 5'-TTTCTAGATTAAGCTGGGGTACTCTATCGTGGT-3'). PCR products were cloned into the pCS2+ vector. *En-zfp665ΔC* and *VP16-zfp665ΔC* constructs were generated by cloning the *zfp665ΔC* sequence into an engineered pCS2+ vector containing an N-terminal engrailed repressor domain or VP16 activation domain, respectively.

## mRNA overexpression in zebrafish embryos

Various plasmid DNA templates were linearized with NotI (NEB), followed by *in vitro* synthesis of capped mRNAs by using an mMESSAGE mMACHINE SP6 Transcription Kit (Life Technologies). The *in vitro* synthesized capped mRNAs were purified by passing through a MEGAclear column (Life Technologies), precipitated with ammonium acetate and ethanol, dissolved in nuclease-free water, and then individually microinjected into 1- or 2-cell stage

zebrafish embryos at different concentrations using an IM-300 microinjector (NARISHIGE). For *zfp665*, 5–20 pg of mRNA was used; for *soxB1b*, 20 pg of mRNA was used; for *tbx2* and *foxn1/4b*, 200 pg of mRNA was used; for En-zfp665ΔC, VP16-zfp665ΔC, and zfp665ΔC, 10 pg of mRNA was used. Injection of *gfp* mRNA at corresponding concentrations served as controls for each experiment. To quantify effects of mRNA overexpression, lengths of rhombomere 5 and the angles of the gene expression domains were measured using ImageJ (NIH). The results are presented as hybrid violin/box/dot plots created with the ggstatsplot R package (version 0.5.0, https://cran.r-project.org/web/packages/ggstatsplot/index.html). Two-tailed Student's *t*-test were conducted using Excel (Microsoft), and Welch's one-way ANOVA with Games-Howell post hoc test plus Bonferroni correction were performed using ggstatsplot for multiple comparisons.

## Supporting information

**S1 Fig. qPCR analyses of known germ granule-enriched transcripts.** Heatmaps show the relative fold changes between two samples from the same embryo for the indicated transcripts. The relative fold changes were normalized to β-actin. The names of characterized transcripts or their IDs of the gene models are shown at the bottom of the corresponding column. The embryo sets selected for RNA sequencing are indicated on the left side of the corresponding rows with different colors. The "no change" group represents embryos in which significant enrichment of germ granule-enriched transcripts could not be detected between the two separated blastomeres. The "uncertain" group indicates that the fold changes of characterized transcripts did not show the expected pattern. The "ΔCt of β-actin > 2" group includes samples that show disparities in the amount of cDNA between samples isolated from the same embryo, suggesting that RNA quality of one of the two paired samples was low. The "opposite" group indicates that the animal and vegetal tiers distinguished by morphology displayed an opposite pattern for fold changes of characterized transcripts. The red to blue color scale indicates relative fold changes between two samples from the same embryo. The red to green color scale indicates ΔCt values of β-actin between two samples from the same embryo. The presumptive germ granule-positive (p), germ granule-negative (n), animal-tier (a), and vegetal-tier (v) blastomeres are indicated.
(TIF)

**S2 Fig. Sequencing depth analyses.** (A) Histograms showing the percentages of mapped and unmapped reads across twelve 2-cell stage or 8-cell stage samples. Identification of samples and their batch of origin are shown at the bottom of the histograms. (B) The pie chart shows the average ratios of mapped and unmapped reads. On average, 32% and 63% of reads were mapped to mitochondrial DNA (green) and chromosomal DNA (yellow), respectively, and 5% of reads could not be mapped (orange) to the reference sequence of *B. floridae*. (C) Distributions of average FPKM values for the 23,517 genes with detectable transcript levels. Among them, 12,445 genes showed moderate-to-high transcript levels (FPKM > 1). (D-E) Saturation curve analyses showed sequencing depths based on million counts over the number of genes with FPKM > 1 across 2-cell stage (D) and 8-cell stage (E) samples. Underlying data are available in S1 Data.
(TIF)

**S3 Fig. The dispersion trend of the RNA-seq datasets.** (A-B) Boxplot of Cook's distances across the twelve 2-cell stage (A) and the twelve 8-cell stage (B) samples showed that there were no detectable outliers among samples. (C-D) The scatter plots show the estimated dispersion values over the mean of normalized counts for each gene in 2-cell stage (C) and 8-cell

stage (D) samples. Black dots denote the dispersion values of each gene analyzed by maximum likelihood estimation (MLE) among the six biological replicates (12 samples). The red line denotes a common trend of dispersion for all samples. Blue dots denote final adjusted dispersion values of each gene analyzed by maximum a posteriori (MAP) estimation; the outliers are labeled with black dots surrounded by blue circles.
(TIF)

**S4 Fig. Visualization of the transcriptome data showing the asymmetric distribution of transcripts based on edgeR analysis.** (A) All symbols are the same as those described in Fig 3. Blue circles (total number = 318) are DETs in the animal tier; red circles (188) are DETs in the vegetal tier; yellow circles (84) are DETs in germ granule-positive blastomere; green circles (15) are DETs in germ granule-negative blastomere; orange circles (60) are DETs in both the germ granule-positive blastomere and the vegetal tier; purple circles (5) are DETs in both the germ granule-negative blastomere and the animal tier. Transcripts that are not enriched are in gray. (B) A scatter plot showing the spatial distributions of nine previously characterized transcripts. (C) Summary information for the nine previously characterized genes. GP, germ granule-positive blastomere. ID, gene model ID from genome assembly v2.0.
(TIF)

**S5 Fig. Zebrafish orthologs of newly identified amphioxus germ granule-enriched transcripts are not aggregated into compact granules in the zebrafish embryo.** Whole-mount *in situ* hybridization of *vasa*, *pomp*, *thoc3*, *samh1*, *kif4*, *cp072* and *kanl2* in 4- to 8-cell stage (A) and 30 hpf (B) zebrafish embryos. The high-magnification views of the expected germ granule region (A) and developing gonad (B) are shown on the right-hand side of each panel. The arrows indicate signals of *vasa* transcripts. The data represent the expression patterns of most samples (>95%, n ≈ 30).
(TIF)

**S6 Fig. Assignment of the animal-vegetal polarity in the 2-cell stage amphioxus embryo by the position of the polar body.** WMISH of three animal pole-localized transcripts at the 2-cell stage. The arrowheads indicate polar bodies visualized by DNA staining with Hoechst.
(TIF)

**S7 Fig. Injection of 20 pg amphioxus *soxB1b* mRNA resulted in the loss of one (22%) or both (19%) eyes in zebrafish embryos.**
(TIF)

**S8 Fig. Differential expression analysis between the 2- and 8-cell stages.** (A-B) MA-plots showing the log2 fold changes between the 2- and 8-cell stage embryos over the average transcript level of each gene by using DESeq2 (A) or edgeR (B). Transcripts showing significantly differential levels (FDR < 0.1) are in red. (C-D) Venn diagrams show the number of transcripts with higher level at the 8-cell stage (8-cell > 2-cell), comparing between DESeq2 and edgeR analyses (C), and the number of such transcripts with or without asymmetric localizations at the 8-cell stage (D). (E) Top 10 biological function GO groups of the transcripts showing higher levels at the 8-cell stage. (F-G) Venn diagrams show the number of transcripts with lower level at the 8-cell stage (8-cell < 2-cell), comparing between DESeq2 and edgeR analyses (F), and the number of such transcripts with or without asymmetric localization at the 8-cell stage (G). (H) Top 10 biological function GO groups of the transcripts showing lower levels at the 8-cell stage. For each group, the GO term with smallest *p*-value is shown. The asterisks indicate biological functions discussed in the text. Underlying data are available in S1 Data.
(TIF)

**S1 Table. The result table from DESeq2 or edgeR analysis.**
(XLSX)

**S2 Table. List of selected DETs for WMISH.**
(XLSX)

**S1 Data. Data underlying Figs 2C, 2F, 2G, 2H, 2I, 2J, 7B, 7D, 7F, 7H, 8A, 8B, 8C, 8E, 8F, 9D, 9F, S2A Fig, S2C Fig, S8C Fig, S8D Fig, S8E Fig, S8F Fig, S8G Fig and S8H Fig.**
(XLSX)

**S2 Data. 2D scatter plot of DETs based on DESeq2.**
(HTML)

**S3 Data. 2D scatter plot of DETs based on edgeR.**
(HTML)

## Acknowledgments

The authors thank Che-Huang Tung, Tzu-Kai Huang, Ching-Yi Lin and the staff at the core facility and the Marine Research Station of the Institute of Cellular and Organismic Biology, and NGS Genomics core facility of the Biodiversity Research Center, Academia Sinica for technical assistance. The authors also thank Marcus Calkins for English editing.

## Author Contributions

**Conceptualization:** Che-Yi Lin, Yi-Hsien Su, Jr-Kai Yu.

**Data curation:** Che-Yi Lin.

**Formal analysis:** Che-Yi Lin.

**Funding acquisition:** Yi-Hsien Su, Jr-Kai Yu.

**Investigation:** Che-Yi Lin, Kun-Lung Li, Yann Le Pétillon, Luok Wen Yong.

**Methodology:** Che-Yi Lin, Mei-Yeh Jade Lu.

**Project administration:** Yi-Hsien Su, Jr-Kai Yu.

**Resources:** Mei-Yeh Jade Lu, Jia-Xing Yue, Yi-Hua Chen, Sheng-Ping L. Hwang.

**Software:** Che-Yi Lin, Jia-Xing Yue.

**Supervision:** Yi-Hsien Su, Jr-Kai Yu.

**Validation:** Che-Yi Lin, Fu-Yu Tsai, Yu-Feng Lyu, Cheng-Yi Chen.

**Visualization:** Che-Yi Lin.

**Writing – original draft:** Che-Yi Lin, Yi-Hsien Su, Jr-Kai Yu.

**Writing – review & editing:** Che-Yi Lin, Mei-Yeh Jade Lu, Jia-Xing Yue, Kun-Lung Li, Yann Le Pétillon, Cheng-Yi Chen, Sheng-Ping L. Hwang, Yi-Hsien Su, Jr-Kai Yu.

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
