## [Decision Letter · Decision Letter 0]

17 Jun 2020

Dear Dr Yu,

Thank you very much for submitting your Research Article entitled 'Molecular asymmetry in the cephalochordate embryo revealed by single-blastomere transcriptome profiling' to PLOS Genetics. Your manuscript was fully evaluated at the editorial level and by independent peer reviewers. The reviewers appreciated the attention to an important problem, but raised some substantial concerns about the current manuscript. As suggested  by the reviewers', the authors should attempt to over-express the genes in amphioxus, which is more relevant than in zebrafish. Further in situ hybridization analysis of the dorsalized phenotype in zebrafish should be performed, as suggested. Analysis of gene expression at the end of gastrulation or early somitogenesis might be most informative. Many additional, valuable comments are provided by the reviewers.  Based on the reviews, we will not be able to accept this version of the manuscript, but we would be willing to review again a much-revised version. We cannot, of course, promise publication at that time.

If you decide to revise the manuscript for further consideration at PLOS Genetics, please aim to resubmit within the next 90 days, unless it will take extra time to address the concerns of the reviewers, in which case we would appreciate an expected resubmission date by email to plosgenetics@plos.org.

[LINK]

We are sorry that we cannot be more positive about your manuscript at this stage. Please do not hesitate to contact us if you have any concerns or questions.

Yours sincerely,

Mary C. Mullins

Associate Editor

PLOS Genetics

Gregory Barsh

Editor-in-Chief

PLOS Genetics

Reviewer's Responses to Questions

**Comments to the Authors:**

Reviewer #1: This manuscript describes single cell RNA-seq results with the aim to list the localized maternal mRNAs in cephalochordate embryos. The authors sequenced cDNA of isolated blastomeres of the 2-cell embryos and cDNA of separated animal and vegetal hemispheres at the 8-cell stage in Branchyostoma. The data was analyzed in pairwise way. Branchyostoma embryo has a single cluster of germ plasm that is partitioned into one of 2 cells at the 2-cell stage and then into the vegetal half at the 8-cell stage. The data suggests that 106 RNAs are localized to the germ plasm, and 222 RNAs to the animal hemisphere. They confirm this by in situ hybridization of 13 randomly selected germ plasm RNAs and 8 animal hemisphere RNAs. Germ plasm RNAs showed localization to tiny germ plasm, while animal hemisphere RNAs had rather broad distribution biased to the animal hemisphere. The author carried out GO analysis and functional analysis of the four animal hemisphere RNAs by overexpression in zebrafish embryos. Zfp665 and soxB1b resulted in dorsalization phenotype. Overall the data is clean and well presented. I appreciate comprehensive listing of localized maternal RNAs to the germ plasm in this study. However, there are some issues need to be addressed in the current manuscript.

Major comments:

1. Introduction, first paragraph. The authors mention that maternally localized cytoplasmic determinants encode transcription factors to provide “mosaic” properties for the blastomeres that inherit them, or alternatively, encode signaling molecules to generate “regulative” inductive signals. This sentence is misleading. Mosaic properties are not only accounted by localized transcription factors. It should be related to localization of maternal factors that are unrelated to the discrimination between transcription factors and signaling molecules. So it is inappropriate to relate localized signaling molecules to regulative property.

2. Introduction, second paragraph. The authors should read the following paper and update the descriptions written in this paragraph.

Maternal Huluwa Dictates the Embryonic Body Axis Through β-Catenin in Vertebrates

Lu Yan, Jing Chen, Xuechen Zhu, Jiawei Sun, Xiaotong Wu, Weimin Shen, Weiying Zhang, Qinghua Tao, Anming Meng

Science. 2018 Nov 23;362

doi: 10.1126/science.aat1045.

3. Introduction, third paragraph. The authors may state that cell fates in the animal hemisphere (ectodermal fates) are basic fates without localized determinant, and the vegetal cell fates were modified by the localized maternal factors in tunicates. In ascidians, formation of the animal-vegetal axis and the anterior-posterior axis is totally independent processes.

4. Line 109. It is better to describe the results of the references 12 and 13 in more detail because these references are hardly obtained (downloaded).

5. Line 114. amphioxus blastomeres separated at the 2-cell stage have different developmental capacities, suggesting that the cells are not identical and certain asymmetrically localized maternal transcripts likely participate in specifying cell fates and body axes in early amphioxus embryos [17].

I do not agree with this sentence. In their previous research (reference 17), the autors showed that blastomeres separated at the 2-cell stage develop into twin larvae that have normal morphology. Therefore, localized germ plasm is likely contribute to germ cell formation, but not to specifying cell fates and body axes in amphioxus embryos that show regulative property as reported since many years ago.

6. Line 251. Due to the limitation of the detection method. Is there a possibility that the RNAs are gradually concentrated to germ plasm from the 2- to 8-cell stage?

7. Fig. 4Ci. FoxN transcripts seems to be localized to the vegetal hemisphere in addition to strong localization to the germ plasm. If this is reproducible result, the authors might mention about this.

8. Fig. 5C. I am wondering how the authors recognized the animal and vegetal poles of the two-cell stage embryos? This should be clearly mentioned.

Minor comments:

1. Titles of S1 Data Table. Titles corresponding to Fig. 2H-J are all “transcript list enriched in the germ granule-positive blastomere”. This would be mistake.

2. Line 163. Interestingly, ratios of the reads mapped to the mitochondrial DNA did not show batch effects. I cannot understand this sentence. Be more specific.

3. Fig. S4. Readers would want to know the positions of the genes that is shown in Fig. 3B, namely, Tcf, nodal, SoxB1b etc.

4. Fig. 4Ba. Syptonemal would be Synaptonemal.

5. Do the authors know the distribution of SoxB1b mRNA in zebrafish embryos?

Reviewer #2: In this study the authors apply RNAseq technology to examine the cellular transcriptomes of each cell of the 2-cell embryo and pooled cells of animal or vegetal identity from the 8 cell stage embryo to address the question of whether or not there is prepattern in this amphioxus species, which is thought to develop in a regulative manner. The authors identified markers that were uniquely expressed in one of two cells at the two stage and that showed differential expression between animal and vegetal halves of the embryo at the 8 cell stage. They went on to validate the expression of a subset of these markers in amphioxus embryos and found evidence that not all blastomeres are equivalent in this embryo. They went on to examine whether germ granule markers showed conserved localization to the germ plasm of zebrafish embryos, and performed overexpression analysis of putative patterning molecules to investigate their effects on zebrafish development. Overall the data are clearly presented, and the manuscript is well-written. The authors identified some new germ granule localized markers and differentially expressed genes that should pave the way for future functional analyses. However, there are a few points detailed below that should be addressed.

Major

When is the zygotic genome activated (ZGA)? Some of the transcripts do not appear to be present or not in the granule at in 2 cell stage, but are in the granule at 8 cell stage (Fig. 4). Similarly, Dag is animal, which seems counter intuitive for a granule marker, and in germ granule? Is this enrichment or de novo transcription in the granule? These should be discussed in the context of ZGA, which likely initiates by this time based on earlier work in other amphioxus species. Related to this point, for the 8 cell stage experiments it appears two pools were created corresponding to animal and vegetal blastomeres. While this seems to be effective for distinguishing between animal and vegetal transcript differences, it assumes that all animal and all vegetal blastomeres are equivalent, which is not the case at least for vegetal since one vegetal blastomere is known to harbor the germ granule/aggregate. It is likely that this could/did result in dilution or underrepresentation of transcripts that may be uniquely enriched in the prospective germline cell or other cell types, especially since the PCR amplification likely introduces bias against low abundance transcripts. Nonetheless new granule localized markers were identified.

The authors perform comparative biological analysis by examining the expression patterns of granule enriched markers and potential axis markers in zebrafish. Among the markers examined for germline localization only one, vasa, was localized to the germ plasm of zebrafish. Based on this the authors conclude that germ granule transcripts are unique between species. However, only early stages (prior to genome activation) were examined. Related to the point above, if some of the markers that localized to the granule of 8 cell stage in amphioxus are zygotic products, then they might also only be localized/expressed/enriched in germ cells of zebrafish after genome activation, which occurs at a later stage than examined in this study. It is important to examine comparable stages when determining whether or not transcripts are unique to the germline in this case. Are these genes localized to the germ granule of other amphioxus species?

No numbers are reported for the number of embryos examined and fraction with the pattern shown for the validation of candidates in Figs 4 and 5.

It is unclear why the authors did not attempt to over express or target these genes in amphioxus to determine the developmental consequences. I did not find the overexpression analysis performed in the zebrafish added much to this study or to mechanisms regulating axis patterning. These experiments simply show that proteins that these proteins can dorsalize zebrafish embryos when overexpressed, which was already predictable based on their molecular identity and activities of proteins with similar functional domains in zebrafish. One limitation with these experiments is that there does not appear to be a corresponding zebrafish ortholog expressed at this time so one cannot perform complementary loss of function experiments to ask if the gene activity is required for patterning or not in zebrafish or experiments to determine if the amphioxus protein can rescue loss of the zebrafish protein. Moreover, it doesn’t really address the function of these genes in amphioxus development. For some of the RNAs the authors performed dose response curves, but in others the numbers seem rather low and there is no indication of how many times the experiments were repeated or how much variation there was from one experimental repetition to the next. This information and statistical analysis should be included. In addition for some experiments, for example the zfp665 experiments in Fig. 8 D, the numbers examined are rather low.

Minor

In figure 7 and 8, what was the control for the injection experiments presumably gfp was the control, but the labels just say “control”? If so, please label as gfp and with the concentration injected rather than just control.

How was the RNA precipitated? This should be indicated in the methods section.

Measuring the angle of the expression domain has become common practice for quantifying expansion or reduction of DV patterning marker expression and should be included.

The numbers (fraction similarly affected) are reported on some of the figure panels but not on others. This should be shown consistently throughout the manuscript.

Presumably Figure S4 color code is the same as in Fig. 3 but a key should be included in the figure.

Reviewer #3: In this manuscript Lin and collaborators describe, using a single cell/low cell number RNAseq approach the transcriptomic asymmetry in amphioxus cells at two early developmental stages, the two cell and eight cell stages. The study is correctly performed and the results generated may be very useful for the amphioxus community but also for EvoDevo researchers.

The generated transcriptomic data allowed to identify several hundreds of differentially expressed transcripts (DET) at these two developmental stages, between the two cells (in the 2cell stage) and between the micromers and macromers (in the 8 cell stage). The authors confirmed their results using in situ hybridization of some of the DET and they tested functionally by overexpression of two of them (as well as chimeric constructions of one of them, zpf665) in zebrafish, showing its capacity to dorsalize the zebrafish embryo.

The manuscript is correctly written, it is clear and interesting, but have some comments/criticisms that should be corrected/added before publication. So my advice is “major corrections”.

Comments/criticisms:

- Lines 66-70: “Modern molecular analyses have demonstrated that maternally localized cytoplasmic determinants encode transcription factors to provide “mosaic” properties for the blastomeres that inherit them, or alternatively, encode signaling molecules to generate “regulative” inductive signals that instruct nearby cells.” Please add references

- Lines 173-174: “a total of 23,517 genes had detectable transcript levels”. In the B. floridae genome paper (Putnam et al 2008), the authors found 21,900 coding genes. Please explain how did you find 23517 genes with detectable transcript levels

-While the authors compare the transcriptome between the two cells of the 2cell stage, or the micromers versus macromers in the 8c stage, I suggest to also compare the 2cell stage with the 8cell stage transcriptomes. Is there any transcription between these two early developmental stages? If any, is this transcription also asymmetric?

-I find that the DETs enriched in the germ-granule negative blastomer is very low, and particularly the number of DET transcripts shared between the two methods used. Is there any explanation for that?

-The authors selected several genes for in situ verification of the RNAseq results but they do not detail how did they chose these genes. Why these genes and not others? Why only 13? 13 doesn’t seem a random number. Does it mean that other genes were also used for in situ but the in situ didn’t work? And finally why the authors only performed double in situ with four of them?

-In the case of the in situs performed with genes expressed in the animal tier (germ granule negative) blastomers, the authors did not explain why they chose 10 and not 13 as in the previous case, and again why did they chose these 10.

- Please exchange the order of phrases in lines 250-251 (“In five cases (Fig 4Ce-g and 4Dk-l), the compact signal could not be detected until the 8- cell stage, possibly due to the limitation of the detection method”) and lines 247-249.

-Concerning the functional approaches, I understand that functional experiments in amphioxus are technically challenging, but other laboratories publish overexpression of RNAs, morpholinos, and even knock-out lineages. I understand that knockout is a really difficult and long experiment, but overexpression of the two mRNAs used in zebrafish (and the chimeras) should be possible in amphioxus isn’t it? This would show whether the phenotype observed in zebrafish is conserved.

-Concerning the phenotype of the dorsalized zebrafish embryos (both in figure 7 A and C and figure 8D), it would be useful to add some in situs with marker genes showing such dorsalization. I believe the authors when they say that these animals are dorsalized, but it is much better to show it clearly with in situs.

**Have all data underlying the figures and results presented in the manuscript been provided?**

Reviewer #1: Yes

Reviewer #2: Yes

Reviewer #3: Yes

PLOS authors have the option to publish the peer review history of their article (what does this mean?). If published, this will include your full peer review and any attached files.

Reviewer #1: No

Reviewer #2: No

Reviewer #3: No

---

## [Decision Letter · Decision Letter 1]

29 Oct 2020

Dear Dr Yu,

Thank you very much for submitting your Research Article entitled 'Molecular asymmetry in the cephalochordate embryo revealed by single-blastomere transcriptome profiling' to PLOS Genetics. Your manuscript was fully evaluated at the editorial level and by independent peer reviewers. The reviewers appreciated the revisions made to the manuscript but identified some aspects of the manuscript that should still be improved. Reviewer 1 and 2 make important points regarding changes to the text, and additionally as pointed out by Reviewer 2, it is important to address how zygotic transcription was distinguished from re-adenylation of transcripts that occurs with a subset of transcripts in many animals.

We therefore ask you to modify the manuscript according to the review recommendations before we can consider your manuscript for acceptance. Your revisions should address the specific points made by each reviewer.

[LINK]

Yours sincerely,

Mary C. Mullins

Associate Editor

PLOS Genetics

Gregory Barsh

Editor-in-Chief

PLOS Genetics

Reviewer's Responses to Questions

**Comments to the Authors:**

Reviewer #1: The authors have responded thoroughly and satisfactorily to my concerns and most of the others comments on the earlier submission. The currently submission is greatly improve over the original submission.

I have three trivial comments.

1. line 102. The sentence would be “A lack of strong anterior determinants would be consistent with the observation that removal of the anterior cytoplasm has no effect on tunicate embryogenesis [10].”

2. line 106. The sentence would be “while the lack of posterior-vegetal determinants in the anterior blastomeres results in default anterior cell fate specification [11]." Please note that the anterior-posterior axis is defined perpendicular to the animal-vegetal axis in ascidian embryos. There should be vegetal hemisphere determinant in ascidian eggs, but the identity of it is still elusive. Presence of animal hemisphere determinants is also unclear. So far the posterior-vegetal determinants have been identified. 

3. line 118. flatted should be replaced with flattened.

Reviewer #2: This is a revised manuscript reporting results from RNAseq studies to examine the cellular transcriptomes of each cell of the 2-cell embryo and pooled cells of animal or vegetal identity from the 8 cell stage embryo to address the question of whether or not there is prepattern in this amphioxus species, which is thought to develop in a regulative manner. The authors identified markers that were uniquely expressed in one of two cells at the two stage and that showed differential expression between animal and vegetal halves of the embryo at the 8 cell stage. They went on to validate the expression of a subset of these markers in amphioxus embryos and found evidence that not all blastomeres are equivalent in this embryo. To explore potential conserved gene functions, they examined whether germ granule markers showed conserved localization to the germ plasm of zebrafish embryos, and performed overexpression assays to investigate the function/activity of putative patterning molecules on zebrafish development. Overall the data are clearly presented, and the manuscript is well-written. The authors identified some new germ granule localized markers and differentially expressed genes that should pave the way for future functional analyses. In this revision they also report evidence suggesting that zygotic transcription begins earlier than previously thought. In this version the authors have addressed my previous concerns; however, a few points remain to be addressed or clarified.

1) The authors report that zygotic transcription begins much earlier than previously expected. This is an interesting and important addition to the manuscript. Because early development, prior to genome activation, is regulated post-transcriptionally, the authors should indicate how they distinguished between de novo transcription and changes in polyadenylation.

2) I understand that the functional studies using amphioxus embryos are challenging and appreciate the authors efforts to attempt to assess gene function in amphioxus. I also understand that because this proved difficult the authors had to rely on functional assays using zebrafish embryos to explore potential activities of their candidate patterning genes. To better distinguish between actual requirements/roles in patterning in this case and potential gene functions or activities, which is what the authors really show, it seems more appropriate to replace “regulates” in the section titled “Amphioxus zfp665 represses..”with “disrupts” or something similar. e.g. (pg. 17 line 395) “to further understand how amphioxus zfp665 regulates axial patterning…” would become “to further understand how amphioxus zfp665 disrupts axial patterning…” and later (pg. 17 line 402) “suggest that zfp665 regulates/…” would become instead “suggest that zfp665 disrupts or perturbs…..” (pg. 18 line 418) “to regulate zebrafish dorsoventral patterning” would be “to disrupt or interfere with zebrafish dorsoventral patterning”

Reviewer #3: Most of my comments and criticisms have been answered correctly. In addition, the authors have added some analyses on the RNAseq samples I requested and the supplementary results are of great interest.

For these reasons, and despite not having been able to perform the over-expression experiments on amphioxus embryos, for reasons I can understand, I accept the manuscript for publication in PloS Genetics

**Have all data underlying the figures and results presented in the manuscript been provided?**

Reviewer #1: Yes

Reviewer #2: Yes

Reviewer #3: Yes

PLOS authors have the option to publish the peer review history of their article (what does this mean?). If published, this will include your full peer review and any attached files.

Reviewer #1: No

Reviewer #2: No

Reviewer #3: No

---

## [Editor Report · Decision Letter 2]

24 Nov 2020

Dear Dr Yu,

We are pleased to inform you that your manuscript entitled "Molecular asymmetry in the cephalochordate embryo revealed by single-blastomere transcriptome profiling" has been editorially accepted for publication in PLOS Genetics. Congratulations!

Yours sincerely,

Mary C. Mullins

Associate Editor

PLOS Genetics

Gregory Barsh

Editor-in-Chief

PLOS Genetics

Comments from the reviewers (if applicable):

**Data Deposition**

http://datadryad.org/submit?journalID=pgenetics&manu=PGENETICS-D-20-00766R2

**Press Queries**

---

## [Editor Report · Acceptance letter]

29 Dec 2020

PGENETICS-D-20-00766R2 

Molecular asymmetry in the cephalochordate embryo revealed by single-blastomere transcriptome profiling 

Dear Dr Yu, 

We are pleased to inform you that your manuscript entitled "Molecular asymmetry in the cephalochordate embryo revealed by single-blastomere transcriptome profiling" has been formally accepted for publication in PLOS Genetics! Your manuscript is now with our production department and you will be notified of the publication date in due course.

With kind regards,

Melanie Wincott

PLOS Genetics

On behalf of:
